## RESEARCH ARTICLE

# Arsenic binds to nuclear transport factors and disrupts nucleocytoplasmic transport

Emma Lorentzon[1], Jongmin Lee[2], Jakub Masaryk[1], Katharina Keuenhof[1], Nora Karlsson[1], Charlotte Galipaud[1], Rebecca Madsen[1], Johanna L. Höög[1], David E. Levin[2] and Markus J. Tamás[1,*]

## ABSTRACT

Human exposure to arsenicals is associated with devastating diseases such as cancer and neurodegeneration. At the same time, arsenic-based drugs are used as therapeutic agents. The ability of arsenic to directly bind to proteins is correlated with its toxic and therapeutic effects, highlighting the importance of elucidating arsenic–protein interactions. In this study, we took a proteomic approach and identified 174 proteins that bind to arsenic in *Saccharomyces cerevisiae*. Proteins involved in nucleocytoplasmic transport were markedly enriched among the arsenic-binding proteins, and we demonstrate that arsenic binding to nuclear import factors results in their relocation from the nuclear envelope and subsequent aggregation in the cytosol. Similarly, nuclear pore proteins that make up the nuclear pore complex mislocalized and aggregated in arsenic-exposed cells. Consequently, arsenic was shown to inhibit nuclear protein import and export. We propose a model in which arsenic binding to nuclear transport factors leads to their mislocalization and aggregation, which disrupts nucleocytoplasmic transport and causes arsenic sensitivity.

KEY WORDS: Arsenic, Karyopherin, Nuclear pore protein, Nuclear pore complex, Nuclear import, Nuclear export, Nuclear envelope

## INTRODUCTION

Human exposure to the poisonous metalloid arsenic is a global health threat that affects hundreds of millions of people (Chen and Costa, 2021). High concentrations of arsenic in the groundwater have been measured in a large number of countries and long-term exposure is associated with numerous human health problems, such as skin disorders, cardiovascular disease, diabetes, cancers of the liver, lung and kidneys, and neurological and neurodegenerative disorders (Chen and Costa, 2021; Rahman et al., 2021; Wysocki et al., 2023). At the same time, arsenic-containing compounds are currently used in anticancer and antiparasitic therapy (Paul et al., 2023).

Pentavalent arsenate, As(V), and trivalent arsenite, As(III), are the most common forms of inorganic arsenic in the environment

(Chen and Costa, 2021). Once inside cells, inorganic arsenic can be enzymatically converted into mono-, di-, and trimethylated metabolites (Thomas, 2021). Various forms of arsenic affect cells and living organisms in distinct ways. As(V) competes with phosphate in biochemical reactions and disrupts adenosine triphosphate (ATP) production. As(III) has a high affinity for sulfhydryl groups, such as the thiol groups of cysteine residues, and binding of As(III) and its metabolites to proteins can disrupt protein conformation, function and interactions (Chen et al., 2019; Kitchin and Wallace, 2008; Shen et al., 2013; Tamás et al., 2014; Vergara-Gerónimo et al., 2021; Wysocki et al., 2023). Methylation affects the toxicity of arsenicals, as well as their modes of action and protein binding specificities (Shen et al., 2013; Thomas, 2021). For example, As(III) can bind up to three cysteine residues, monomethylarsenite [MAs(III)] can bind two cysteine residues, and dimethylarsenite [DMAs(III)] can bind only one cysteine residue (Shen et al., 2013). The ability of arsenic to bind to proteins is associated with its toxicity, but also with its therapeutic effects. For example, binding to cysteine residues in the oncoprotein PML-RARα underlies the anticancer activity of arsenic trioxide in individuals with acute promyelocytic leukemia (APL) (Lallemand-Breitenbach et al., 2008; Zhang et al., 2010). Similarly, arsenic-binding to specific kinases and transcriptional regulators is linked to arsenic resistance in yeast and bacteria (Guerra-Moreno et al., 2019; Kumar et al., 2016; Shi et al., 1994). Although the toxicity of trivalent arsenite has traditionally been attributed to its interactions with sulfhydryl groups in native (folded) proteins (Kitchin and Wallace, 2008; Shen et al., 2013), more recent studies have shown that As(III) also targets non-native proteins and impairs their proper folding (Hua et al., 2022; Jacobson et al., 2012; Ramadan et al., 2009; Sapra et al., 2015). In cells, this results in extensive protein misfolding and aggregation that, in turn, has a negative effect on cell proliferation and viability (Andersson et al., 2021; Hua et al., 2022; Ibstedt et al., 2014; Jacobson et al., 2012).

Thus, knowledge of arsenic–protein interactions is key to understanding the toxic and therapeutic effects of arsenicals, as well as cellular sensing and defense mechanisms. Several large-scale studies have been performed with the aim of identifying arsenic-binding proteins (Liu et al., 2023). For example, 360 proteins bound to arsenic *in vitro* using a human proteome microarray (Zhang et al., 2015a) and *in vivo* studies identified 40 arsenic-binding proteins in APL cells (Zhang et al., 2015b), 50 proteins in human breast cancer cells (MCF-7 cell line) (Zhang et al., 2007), 51 proteins in human embryonic kidney epithelial cells (HEK293T) (Dong et al., 2022) and 48 proteins in A549 human lung carcinoma cells (Yan et al., 2016). Follow-up experiments indicated that some of these proteins are bona fide targets of arsenic binding and inhibition (Yan et al., 2016; Zhang et al., 2015a,b). Although identifying the arsenic-binding proteome is a promising approach to addressing toxicity and resistance mechanisms, the aforementioned *in vivo* studies identified relatively few targets. Thus, a comprehensive catalogue of *in vivo* arsenic–

[1]Department of Chemistry and Molecular Biology, University of Gothenburg, Box 462, S-405 30 Göteborg, Sweden. [2]Department of Molecular and Cell Biology, Boston University Henry M. Goldman School of Dental Medicine, Boston, MA 02118, USA.

*Author for correspondence (markus.tamas@cmb.gu.se)

J.L.H., 0000-0003-2162-3816; D.E.L., 0000-0003-0696-2860; M.J.T., 0000-0002-0762-7848

protein interactions and the resulting consequences on cell physiology is still lacking.

In this study, we took a proteomic approach and identified 174 proteins that bind to arsenic in the budding yeast *Saccharomyces cerevisiae*. Proteins involved in nucleocytoplasmic transport were strongly enriched among the arsenic-binding proteins, and data from follow-up experiments suggest that arsenic binding to nuclear transport factors leads to their mislocalization and aggregation, thereby disrupting protein transport across the nuclear envelope and causing arsenic sensitivity.

## RESULTS

### Proteome-wide identification of arsenic-binding proteins in yeast

To identify proteins that bind to arsenic *in vivo*, we took an unbiased proteomic approach in *S. cerevisiae* using biotin-conjugated As(III) (hereafter As–biotin) as a probe (Kumar et al., 2016; Lee and Levin, 2019). We previously noted that As–biotin cannot discriminate between As(III)-binding proteins and proteins that bind to MAs(III), owing to intracellular conversion of As(III) into MAs(III) (Lee and Levin, 2018, 2019). Therefore, we incubated yeast cells that lack the methyltransferase Mtq2 responsible for As(III) methylation (Lee and Levin, 2018) with 50 µM As–biotin without or with a 10 min pre-treatment with 1 mM As(III) or 500 µM MAs(III) as blocking agents (Fig. 1A). The pretreatments were performed to obtain an indication of which arsenical binds to each protein, given that binding of As–biotin to a protein is expected to be attenuated in the presence of As(III) or MAs(III); the *mtq2Δ* cells were used to reduce metabolism of the blocking arsenical, and a lower concentration of MAs(III) was used because MAs(III) is more toxic than As(III) (Kumar et al., 2016; Lee and Levin, 2018, 2019, 2022). After cell disruption and As–biotin pull-down with streptavidin-agarose beads, candidate arsenic-binding proteins were eluted, separated by SDS-PAGE and identified by microcapillary liquid chromatograph-tandem mass spectrometry (LC-MS/MS) (Fig. 1A). As a control, we performed pull-downs using cells that had not been incubated with As-biotin. In total, 776 proteins were identified in at least one of the conditions (Table S1). To select candidate arsenic-binding proteins, we filtered the 776 proteins using the following criteria: (1) no peptide present in the control, (2) ≥5 unique peptides identified after As-biotin pull-down and (3) ≥2-fold reduction of signal intensity when competitor As(III) or MAs(III) was present during pull-down. Applying these stringent filtering criteria provided a list of 174 proteins (Table S2), which, to our knowledge, represents the largest set of *in vivo* arsenic-binding proteins reported to date.

Several of the 174 yeast proteins have human orthologues that have previously been reported to bind to arsenic in large-scale *in vitro* and *in vivo* screens (Dong et al., 2022; Zhang et al., 2015a, 2007), including: subunits of the chaperonin T-complex protein ring (TRiC), also known as chaperonin containing TCP-1 (CCT) complex (hereafter TRiC/CCT) involved in protein folding; metabolic enzymes such as glycerol-3-phosphate dehydrogenase, aldehyde dehydrogenase and members of the pyruvate dehydrogenase complex; proteins involved in DNA replication, including components of the minichromosome maintenance (MCM) complex; α- and β-tubulin; and ribonucleotide reductase, which is implicated in DNA synthesis and repair. Thus, these proteins may represent evolutionarily conserved arsenic-binding targets. Of these, the chaperonin TRiC/CCT (Pan et al., 2010), tubulin (Zhang et al., 2007) and pyruvate dehydrogenase (Bergquist et al., 2009; Peters et al., 1946) have been proposed to be direct toxicity targets.

The majority of the 174 proteins (103 proteins, 59%) had ≥2-fold reduced signal intensity in the presence of As(III) as well as MAs(III), suggesting that they may bind both arsenicals (Table S2). Forty-eight proteins (28%) reached the threshold of ≥2-fold reduction in signal intensity only in the presence of MAs(III), while 23 (13%) reached the threshold only in the presence of As(III), suggesting that these proteins preferentially bind to either MAs(III) or As(III), respectively. As(III) and MAs(III) preferentially bind to the thiol group of cysteine residues in proteins (Kitchin and Wallace, 2008; Shen et al., 2013; Zhang et al., 2015a), and virtually all 174 proteins (99.4%) contained at least one cysteine compared to 90.7% in the yeast proteome ($P<10^{-7}$) (Fig. 1B). The arsenic-binding set was also significantly enriched for proteins with cysteines adjacent or proximal to other cysteines (CC, CxC, CxxC and CxxxC motifs), and the mean number of cysteines and CC motifs per protein was significantly higher in arsenic-binding proteins compared to the proteome (Fig. 1B). Additionally, we observed a significant overlap between the arsenic-binding proteins and a set of 145 yeast proteins that possess surface-exposed reactive cysteines (12 proteins, $P=0.0002$) (Marino et al., 2010) (Fig. 1C). As(III) has been shown to bind to proteins containing zinc finger motifs, specifically to C3H1 and C4 motifs (Vergara-Gerónimo et al., 2021; Zhou et al., 2011). Twelve of the 174 proteins ($P=0.11$) in our dataset are putative zinc-binding proteins, of which eight are predicted to contain C3H1 and C4 motifs (Wang et al., 2018). In summary, the As-biotin probe identified proteins that bind to As(III) or MAs(III), or both arsenicals, and our findings reinforce the strong preference of As(III) and MAs(III) for cysteine residues in proteins *in vivo*.

### Protein binding as a possible toxicity mechanism

It has been postulated that trivalent arsenic causes toxicity via protein binding, which inactivates or depletes important cellular functions (Chen et al., 2019; Kitchin and Wallace, 2008; Shen et al., 2013; Tamás et al., 2014; Vergara-Gerónimo et al., 2021; Wysocki et al., 2023). However, only few direct toxicity targets and mechanisms have been described to date. One way to pinpoint candidate toxicity targets is to perform drug-induced haploinsufficiency profiling (HIP) assays (Giaever et al., 1999; Lum et al., 2004). A previous yeast HIP study identified 33 As(III)-sensitive heterozygous diploid knockout mutants (Pan et al., 2010), of which eight encode proteins that were present in the arsenic-binding set ($P<10^{-6}$) (Fig. 1D), including components of the TRiC/CCT complex [Cct1 (also known as Tcp1), Cct4, Cct5 and Cct7], α-tubulin (Tub3), the nuclear pore protein Nup145, adenine-requiring Ade12 and the serine palmitoyltransferase Lcb1. These proteins might represent bona fide arsenic toxicity targets, as proposed for TRiC/CCT (Pan et al., 2010) and tubulin (Zhang et al., 2007).

Another way to identify toxicity targets is to integrate chemical-genetic and genetic interaction data (Parsons et al., 2004). For this, we retrieved negative genetic interactors of selected arsenic-binding proteins and asked whether the sets of negative genetic interactors are enriched for As(III)-sensitive mutants. Indeed, we observed significant enrichments in As(III) sensitivity among negative genetic interactors of selected arsenic-binding protein-encoding genes involved in transport across the nuclear envelope [*KAP121* (also known as *PSE1*) ($P=0.0004$), *NUP84* ($P<10^{-20}$)], components of TRiC/CCT [*CCT1* ($P<10^{-13}$), *CCT5* ($P<10^{-7}$)], the α-subunit of pyruvate dehydrogenase *PDA1* ($P<10^{-13}$), the translation regulator *GCN20* ($P=0.004$) and the serine palmitoyltransferase *LCB1* ($P=0.0002$) (Fig. S1). Thus, the tested arsenic-binding proteins might represent bona fide toxicity targets, as proposed for TRiC/

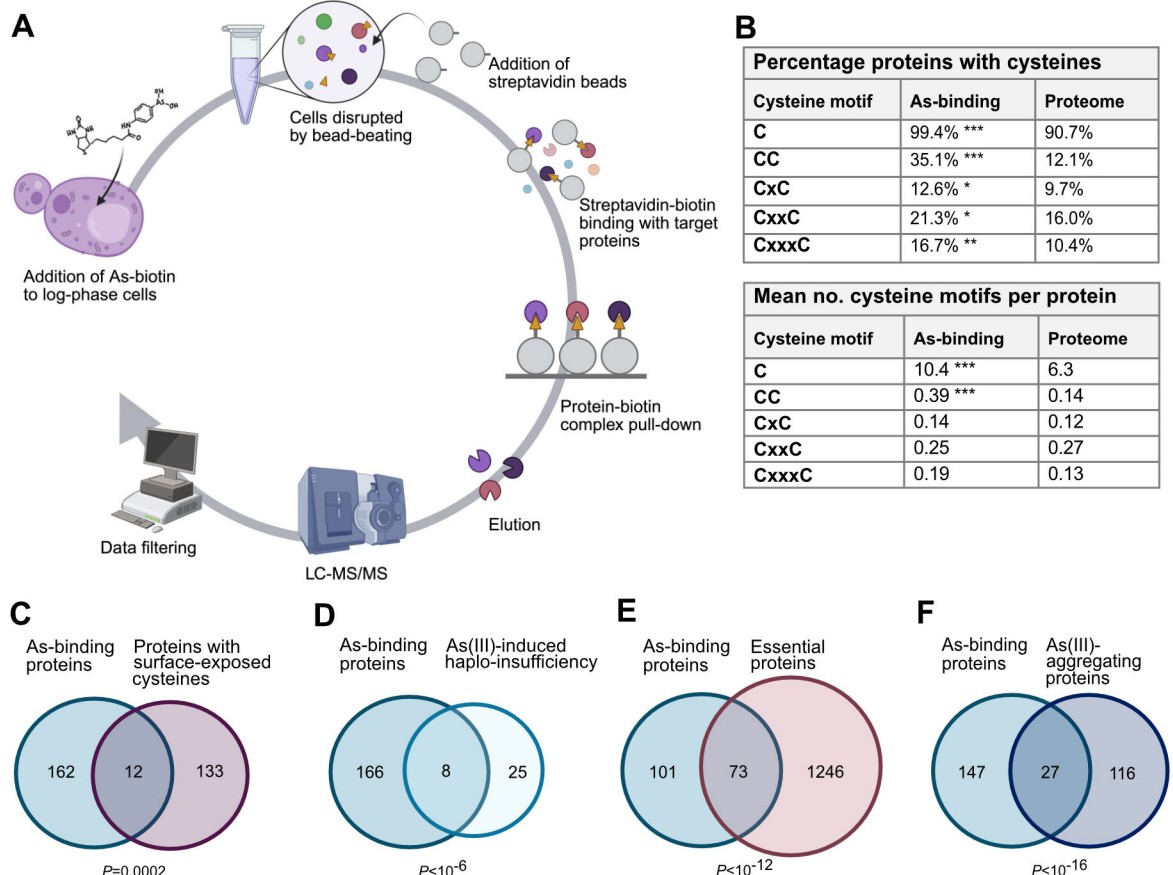

**Fig. 1. A proteome-wide screen identifies arsenic-binding proteins and toxicity targets.** (A) Workflow. Yeast cells (*mtq2Δ*) were incubated with 50 µM As–biotin for 10 min without or with a 10 min pre-treatment with 1 mM As(III) or 500 µM MAs(III) as blocking agents. After cell disruption and protein pull-down using streptavidin beads, the proteins present in the pull-down were identified using LC-MS/MS. The data were filtered using the following stringent criteria: (1) no peptide present in the control; (2) ≥5 unique peptides identified per protein after As–biotin pull-down; and (3) ≥2-fold reduction of signal intensity when competitor As(III) or MAs(III) was present during pull-down. A total of 174 candidate arsenic-binding proteins were identified. (B) Cysteine content and motifs in arsenic-binding proteins versus a proteome of around 5800 proteins (Ho et al., 2018). x represents any amino acid present between the cysteine residues in a motif. Significance was calculated using the hyper-geometric test: *$P<0.05$, **$P<0.01$ and ***$P<0.001$. (C-F) Venn diagrams show the overlap between arsenic-binding proteins and (C) proteins containing surface-exposed cysteine residues (Marino et al., 2010), (D) As(III)-sensitive heterozygous diploid knockout mutants (Pan et al., 2010), (E) essential proteins in *S. cerevisiae* (extracted from SGD; Wong et al., 2023) and (F) proteins aggregating during As(III) exposure (Ibstedt et al., 2014; Jacobson et al., 2012). The significance of the overlaps between the datasets was calculated by the hyper-geometric test and the corresponding *P*-values are indicated.

CCT (Pan et al., 2010) and pyruvate dehydrogenase (Bergquist et al., 2009; Peters et al., 1946).

We noted that a substantial fraction of the arsenic-binding proteins is essential for cell viability (73 proteins, 42%, $P<10^{-12}$) (Fig. 1E), suggesting that arsenic binding might drain the active pool of these essential proteins, resulting in poor growth or survival of yeast cells during arsenic stress. The set of essential arsenic-binding proteins included proposed toxicity targets, such as TRiC/CCT (Pan et al., 2010) and tubulin (Zhang et al., 2007), and previously unreported candidate targets, such as proteins involved in nucleocytoplasmic transport (Fig. S1).

One way arsenic binding affects protein function is by interfering with their folding, thereby preventing proteins from reaching their native fold and, hence, their active conformation (Andersson et al., 2021; Hua et al., 2022; Ibstedt et al., 2014; Jacobson et al., 2012). We found a significant overlap between the set of arsenic-binding proteins and a set of 143 proteins that aggregated in As(III)-exposed yeast cells (27 proteins, $P<10^{-16}$) (Ibstedt et al., 2014; Jacobson et al., 2012) (Fig. 1F), suggesting that arsenic binding to these proteins results in their misfolding and aggregation.

Taken together, our findings suggest that a large fraction of the 174 arsenic-binding proteins identified here may represent bona fide toxicity targets and they support the notion that protein binding is a major arsenic toxicity mechanism that involves affecting protein folding and/or activity. Thus, integrating arsenic–protein binding data with other datasets is a powerful approach to identify previously unreported toxicity targets.

## Arsenic binds to proteins involved in nucleocytoplasmic transport

We next addressed if specific categories of protein functions are over-represented among the arsenic-binding proteins. Gene ontology (GO) analysis revealed that this set was enriched in processes associated with protein import into the nucleus; chaperonin-mediated protein folding; carboxylic acid metabolic processes; nuclear pore localization and organization; DNA unwinding involved in DNA replication; tRNA transport, methylation and aminoacylation; and sphingosine and long-chain fatty acid metabolism (Fig. 2A). Remarkably, the set of arsenic-binding proteins was markedly enriched for functions in nucleocytoplasmic transport and included

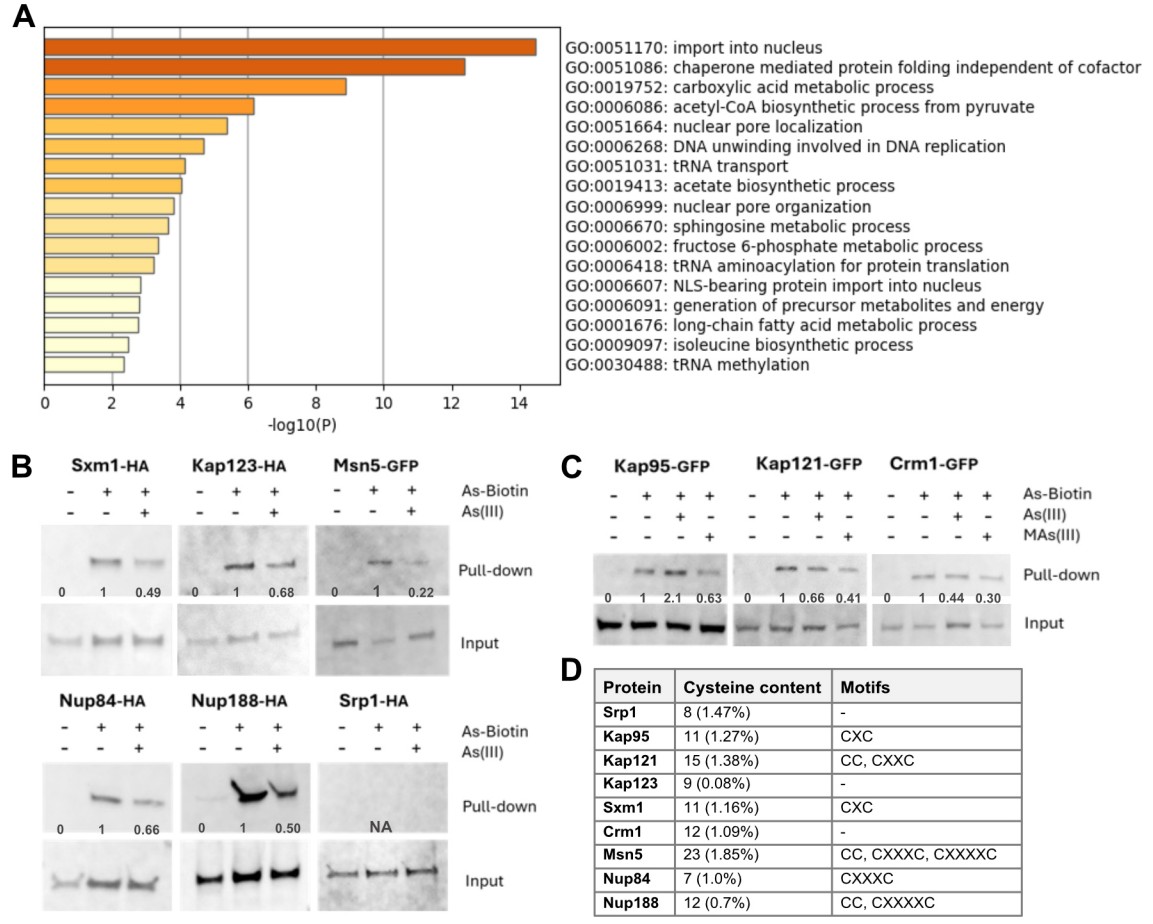

**Fig. 2. Arsenic binds to proteins involved in nucleocytoplasmic transport.** (A) Bar plots of over-represented GO terms in the arsenic-binding protein set using Metascape (Zhou et al., 2019). (B) Cells expressing HA-tagged or GFP-tagged versions of Sxm1, Kap123, Msn5, Nup84, Nup188 and Srp1 were incubated with 50 µM As–biotin for 10 min followed by cell disruption and protein pull-down using streptavidin beads. The proteins were detected by western blot using anti-HA and anti-GFP antibodies. Where indicated, cells were pre-treated for 10 min with 1 mM As(III) as a blocking agent. The loading control (Input) represents 25% of the total lysate. The blots shown are representative of at least two biological repeats, except for Msn5, which was carried out once. Band intensities were quantified (see Materials and Methods) and values are shown below the blots. Pulldown-to-input ratios were calculated for each sample. The ratio for the As–biotin sample was set to 1, and relative values for samples pre-treated with As(III) were obtained by dividing their ratios by the As–biotin ratio. Molecular sizes of the proteins were as follows: Sxm1-HA, 108 kDa; Kap123-HA, 123 kDa; Nup84-HA, 84 kDa; Nup188-HA, 188 kDa; Srp1-HA, 60 kDa; Msn5-GFP, 169 kDa. (C) As–biotin pull-down assays were performed as in B using cells expressing GFP-tagged Kap95, Kap121 and Crm1. The proteins were detected by western blot using an anti-GFP antibody. Where indicated, cells were pre-treated for 10 min with 1 mM As(III) or 500 µM MAs(III) as blocking agents. The blots shown are representative of at least two biological repeats. Band intensities were quantified as in B and values are shown below the blots. Molecular sizes of the proteins are as follows: Kap95-GFP, 122 kDa; Kap121-GFP, 148 kDa; Crm1-GFP, 151 kDa. (D) Cysteine content and motifs in the listed proteins. X represents any amino acid present between the cysteine residues in a motif.

eight out of the 11 importins [Kap104, Kap114, Kap122, Kap123, Kap121, Sxm1 (also known as Kap108), Nmd5 (also known as Kap119) and Mtr10 (also known as Kap111)] present in *S. cerevisiae*, several exportins [Cse1 (also known as Kap109), Crm1 (also known as Kap124) and Msn5 (also known as Kap142)], and numerous nuclear pore proteins (Nup84, Nup85, Nup120, Nup133, Nup145, Nup157, Nup170, Nup188 and Nup192) (Table S2). This strong enrichment, together with our genetic analyses above implicating some of these proteins as direct toxicity targets (Nup145, Nup84 and Kap121), raised the possibility that nucleocytoplasmic transport is a key target of arsenic in living cells.

Importins are nuclear transport factors that bind to cargo proteins containing nuclear localization signals (NLS) in the cytoplasm and facilitate their passage through the nuclear pore complex (NPC) into the nucleus, exportins mediate the export of cargo proteins back to the cytoplasm, while nuclear pore proteins (Nups) constitute the NPC (Aitchison and Rout, 2012; Wing et al., 2022). Importins

and exportins belong to the karyopherin (Kap) family of nuclear transport factors. Although most karyopherins can bind to their cargos directly (karyopherin β), in the case of the heterodimeric α–β complex, it is karyopherin α that binds to the cargo while karyopherin β stabilizes and enhances this interaction (Aitchison and Rout, 2012; Wing et al., 2022). To validate arsenic-binding to selected importins (karyopherin βs Kap123, Kap121 and Sxm1), exportins (karyopherin βs Crm1 and Msn5) and Nups (Nup84 and Nup188), we either introduced plasmids that expressed HA-tagged versions of the corresponding genes into yeast cells or used cells that harbored GFP-tagged versions of the genes in their genomes, and performed As–biotin pulldown assays. All tested proteins bound to As–biotin and this binding was attenuated to varying degrees in the presence of competitor As(III) or MAs(III) (Fig. 2B,C). Thus, these proteins directly bind arsenic *in vivo* in form of As(III) and/or MAs(III). We also tested arsenic binding to the karyopherin α Srp1 (also known as Kap60) and the karyopherin β Kap95, even though

they were not present in the hit list (no peptides were found for Srp1, whereas Kap95 was below the threshold) because these proteins constitute the heterodimeric α/β complex that plays a key role in nuclear transport of NLS-containing proteins in yeast (Aitchison and Rout, 2012). Kap95 readily bound to As–biotin and this binding was attenuated in the presence of MAs(III) (Fig. 2C), suggesting that Kap95 binds to arsenic in form of MAs(III) *in vivo*. Curiously, the presence of As(III) resulted in more Kap95-GFP being pulled-down by As–biotin (Fig. 2C, Table S2), the underlying reason remains unknown (see Discussion). In contrast, Srp1 did not bind to As–biotin (Fig. 2B).

Most of the tested proteins (Sxm1, Kap95, Kap121, Msn5, Nup84 and Nup188) have adjacent or proximal cysteines in their primary sequence (Fig. 2D). Analyses of their known or predicted 3D structures revealed that most of these proteins contain cysteine pairs within ∼5 Å (1 Å=0.1 nm) of each other (Fig. S2), making them suitable substrates for As(III) and/or MAs(III) in their native folded structures. In contrast, Kap123 and Crm1 lack adjacent or proximal cysteines in their primary sequence, raising the question of how As(III) and/or MAs(III) bind to these proteins. Inspection of their 3D structures revealed the presence of proximal cysteines that could potentially serve as binding sites in both proteins (Fig. S2). Finally, Srp1 does not have cysteine motifs in its primary sequence and the closest cysteines in its 3D structure are separated by ∼10 Å with the thiol groups pointing in opposite directions (Fig. S2), explaining why this protein is a poor substrate for As(III) and/or MAs(III).

## Importins mislocalize and aggregate in As(III)-exposed cells

Having established that arsenic binds to individual importins, we next addressed the consequence(s) of this binding. First, we monitored the localization of chromosomally integrated Kap95-GFP. We chose to focus on Kap95 because: (1) it plays a central role in nuclear transport of NLS-containing proteins in yeast (Aitchison and Rout, 2012); (2) integration of chemical-genetic and genetic interaction data suggests that Kap95 may be a direct arsenic toxicity target (Fig. S1); and (3) the human orthologue of Kap95, KPNB1 (also known as importin 90), has been identified as a candidate arsenic-binding protein in MCF-7 cells (Zhang et al., 2007). In untreated (control) cells, chromosomally integrated Kap95-GFP localized around the yeast nuclear envelope (NE) (Fig. 3A,B). In cells exposed to 1.5 mM As(III) for 1 h, NE localization of Kap95-GFP was disrupted and the protein was instead found in distinct foci that were dispersed throughout the cytosol in the majority of cells (Fig. 3A,B). Cytosolic Kap95-GFP foci were also formed when the translation inhibitor cycloheximide (CHX) was added at the same time as As(III) (Fig. 3B), suggesting that As(III) impacts the localization of the native (folded) form of Kap95. As for Kap95, As(III) stress altered the distribution of chromosomally integrated Srp1-GFP from the NE to distinct cytosolic foci (Fig. 3A), both in the absence and presence of CHX (Fig. S3A), which is consistent with the two proteins forming a heterodimer (Aitchison and Rout, 2012).

Controlled formation of protein condensates can be used by cells for various physiological purposes, whereas aggregation of misfolded proteins represents an irreversible loss of protein function (Alberti and Hyman, 2021). To address whether formation of cytosolic Kap95-GFP and Srp1-GFP foci is reversible, we monitored their localization after As(III) exposure for 1 h followed by As(III) washout. NE localization of both proteins slowly recovered after As(III) washout, and this recovery largely coincided with the disappearance of cytosolic Kap95-GFP (Fig. 3C) and Srp1-GFP (Fig. S3B) foci. CHX, added after the washing step, slowed down or prevented the recovery of Kap95-GFP and Srp1-GFP at the NE, as

well as the disappearance of cytosolic Kap95-GFP and Srp1-GFP foci (Fig. 3C, Fig. S3B), indicating that foci reversal and signal recovery at the NE require *de novo* protein synthesis. Thus, formation of Kap95-GFP and Srp1-GFP foci may not be a regulated process that cells use to recover quickly once As(III) stress is relieved.

Next, we addressed whether the Kap95-GFP and Srp1-GFP foci represent aggregated forms of these proteins. For this, we isolated total and aggregated proteins by differential centrifugation and separated the proteins in each fraction by SDS-PAGE followed by immunoblotting with an anti-GFP antibody. Both Kap95-GFP and Srp1-GFP were present in the aggregated protein fractions isolated from As(III)-exposed cells, whereas they were largely absent in the aggregated protein fractions of unexposed cells (Fig. 3D). This finding indicates that Kap95 and Srp1 aggregate in the presence of As(III) and that the cytosolic foci likely represent aggregated forms of these proteins. The data also suggest that arsenic binding to Kap95, in form of MAs(III), is sufficient to induce mislocalization and aggregation of both Srp1 and Kap95 in the heterodimeric α/β complex. Importantly, two additional karyopherin βs, Kap121 (Fig. 3D) and Kap123 (Jacobson et al., 2012), also aggregated during As(III) exposure. We conclude that arsenic binding to nuclear import factors leads to their relocation from the nuclear envelope and subsequent aggregation in the cytosol.

## As(III) affects Nup localization, NE morphology and NPC numbers

The arsenic-binding Nup proteins identified in this study are located in the outer and the inner rings of the NPC (Fig. 4A). Nup145 is, after proteolytic cleavage, present in the outer ring (Nup145C fragment) and in the NPC core (Nup145N) as one of several so-called FG-Nups that regulate selective transport through the NPC (Aitchison and Rout, 2012; Wing et al., 2022). Like the tested importins, As(III) affected the localization of chromosomally integrated Nup84-GFP and Nup188-GFP. In untreated (control) cells, Nup84-GFP and Nup188-GFP localized in patches in the NE (Fig. 4B). During As(III) exposure, both Nup84-GFP and Nup188-GFP were visible as cytosolic foci in a substantial fraction of the cells (Fig. 4B,C). These findings imply that arsenic-binding to Nup84 and Nup188 leads to their mislocalization. CHX did not prevent As(III)-induced Nup84-GFP foci formation (Fig. S4), suggesting that As(III) affects the native (folded) form of Nup84.

We next performed immunoelectron microscopy (EM) on untreated and As(III)-exposed yeast cells using a primary antibody that detects Nups in conjunction with a secondary 10 nm gold label to simultaneously observe NE morphology, Nup localization, NPC morphology and number, and protein aggregates visible as electron-dense content (EDC) within cells (Panagaki et al., 2021; Schneider et al., 2024). Although we did not detect any abnormalities in NPC morphology in As(III)-exposed cells (Fig. 5A), the exposed cells had fewer visible NPCs per cell section compared to unexposed cells (Fig. 5B). The lower number of NPCs is probably not a result of reduced Nup levels during As(III) stress, as Nup84 and Nup188 levels remain unchanged during As(III) exposure (Guerra-Moreno et al., 2015). Instead, a substantial fraction of Nup immunolabelling was associated with EDCs in As(III)-exposed cells (Fig. 5C), suggesting that Nups aggregate. These aggregates were not membrane enclosed but often localized at sites of NE deformations in which the NE extended into the cytoplasm (Fig. 5D, panel I) or where the inner and outer leaflets were separated with the outer leaflet extending into the cytoplasm (Fig. 5D, panel II) forming an outer membrane bud. In summary, arsenic binding to Nups results in their mislocalization and aggregation, reducing the number of NPCs present on the NE.

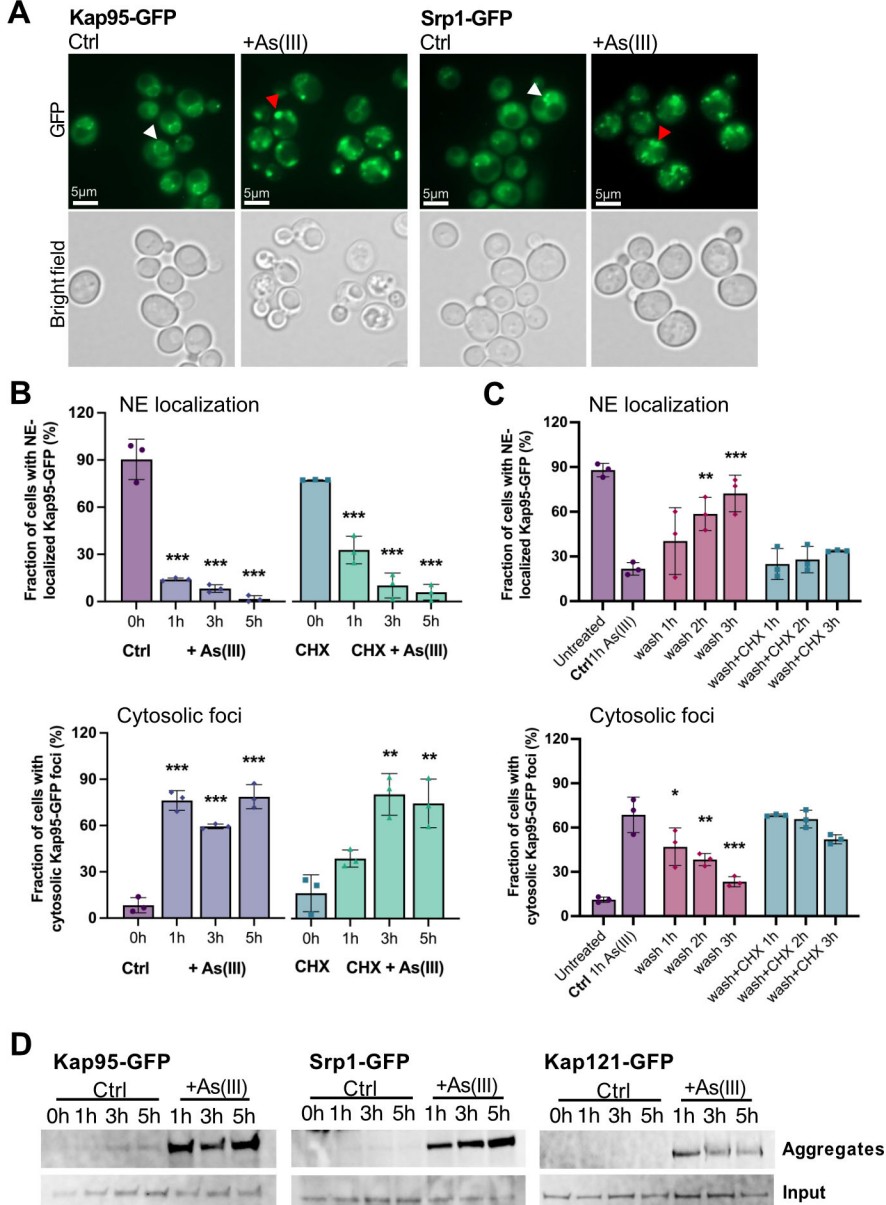

**Fig. 3. Importins mislocalize and aggregate in As(III)-exposed cells.** (A) Localization of GFP-tagged Kap95 and Srp1. In unexposed (Ctrl) cells, the proteins are located around the NE (white arrowheads). After As(III) exposure (1.5 mM, 1 h), the proteins are found in distinct foci dispersed throughout the cytosol (red arrowheads). Images shown are representative of three biological repeats of 100 cells each. (B) Quantification of Kap95-GFP NE localization (top panel) and cytosolic foci formation (bottom panel) in the absence and presence of 1.5 mM As(III) for 1 h and/or 0.2 mg/ml cycloheximide (CHX). Kap95-GFP distribution was scored by fluorescence microscopy and quantified by visual inspection. Data are mean±s.d. of three independent biological repeats of a total of 300 cells. Significance was calculated using an unpaired two-tailed Student's *t*-test with either the untreated control [for As(III)-treated cells] or CHX [for CHX+As(III)-treated cells] as the comparison. \*\**P*<0.01, \*\*\**P*<0.001. (C) Cells were exposed to 1.5 mM As(III) for 1 h, then washed twice and resuspended in medium without As(III) in the presence or absence of 0.2 mg/ml CHX. Kap95-GFP distribution was scored and quantified as in B. Data are mean±s.d. of three independent biological repeats of a total of 300 cells. Significance was calculated using an unpaired two-tailed Student's *t*-test of three independent biological replicates, with 1 h As(III)-exposed cells as the control. \**P*<0.05, \*\**P*<0.01, \*\*\**P*<0.001. (D) Kap95, Srp1 and Kap121 aggregate in the presence of As(III). Cells expressing GFP-tagged proteins were left untreated (Ctrl) or exposed to 1.5 mM As(III) for 1 h, lysed, and the total and aggregated protein fractions were isolated followed by western blot using an anti-GFP antibody. The input (20 µg protein) represents ~1% of the total lysate. The blots shown are representative of at least two biological repeats. Molecular size of the proteins: Kap95-GFP, 122 kDa; Srp1-GFP, 87 kDa; Kap121-GFP, 148 kDa.

## Nuclear transport is disrupted during long-term As(III) stress

Having established that importins and Nups mislocalize and aggregate, we next addressed whether nuclear transport is affected in As(III)-exposed cells using established reporters (Barrientos et al., 2023; Rempel et al., 2019). We first monitored the localization of the Srp1/Kap95 substrate GFP-tcNLS (GFP with a tandem classical NLS) under the control of the galactose-inducible *GAL1* promoter. Expression of GFP-tcNLS was induced and steady-state localization of GFP-tcNLS determined by calculating the ratio of the fluorescence measured in the nucleus over the cytosol (N/C ratio) in the absence or presence of As(III) (Fig. 6A,B). Importantly, the N/C ratio significantly decreased in As(III)-exposed cells (Fig. 6A,B), suggesting that nuclear accumulation of GFP-tcNLS was inhibited. Similarly, the N/C ratios of two other reporters containing NLS sequences recognized, respectively, by Kap104 (Nab2NLS-GFP) and Kap121 (Pho4NLS-GFP), also decreased in As(III)-exposed cells. In contrast, the N/C ratio of GFP without an NLS remained unaffected during exposure (Fig. 6A,B). The total protein levels of Kap95, Srp1 and Kap121

were not affected by As(III) (Fig. 3D). Thus, nuclear import of NLS-containing cargos (GFP-tcNLS, Nab2NLS-GFP, Pho4NLS-GFP) is disrupted in As(III)-exposed cells, possibly due to arsenic binding to import factors and Nups.

To address whether As(III) also affects nuclear export, we monitored the localization of the transcriptional repressor protein Mig1-GFP, which is nuclear in the presence of glucose but rapidly exits the nucleus via Msn5-dependent export when glucose is replaced by glycerol (DeVit and Johnston, 1999). Note that As(III) directly binds to Msn5 (Fig. 2B). As expected, shifting cells from glucose to glycerol resulted in a substantial drop of nuclear Mig1-GFP within 5 min (Fig. 6C,D). When cells were preincubated with As(III) for 1 h, glycerol-stimulated nuclear exit of Mig1-GFP was significantly inhibited (Fig. 6C,D). Thus, As(III) also disrupts nuclear export, possibly by binding to export factors.

Previous work has shown that yeast cells depend on transcriptional regulation of genes required for As(III) tolerance (Wysocki and Tamás, 2010). For example, the transcription factors Yap1 and Msn2 accumulate in the nucleus during As(III) stress, where they induce

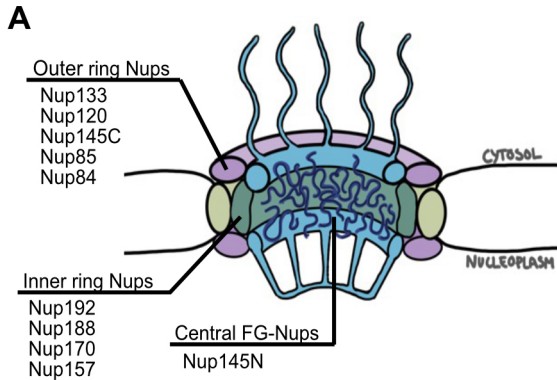

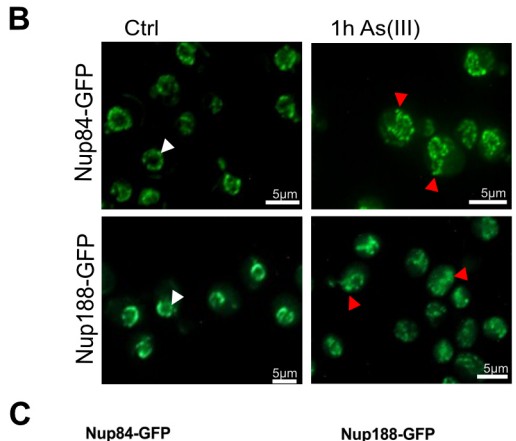

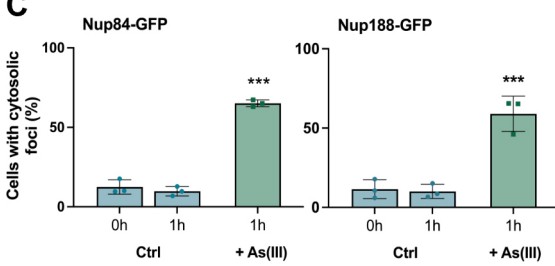

**Fig. 4. As(III) affects Nup localization in cells.** (A) Illustration of the yeast NPC with the arsenic-binding Nups and their respective subcomplex. (B) Localization of GFP-tagged Nup84 and Nup188. In unexposed (Ctrl) cells, the proteins are located around the NE (white arrowheads). After 1 h of As(III) exposure (1.5 mM), the proteins are visible as cytosolic foci (red arrowheads). The images shown are representative of three biological repeats of 100 cells each. (C) Nup84-GFP and Nup188-GFP distribution was scored by fluorescence microscopy and quantified as in Fig. 3B. Data are mean±s.d. of three independent biological repeats of a total of 300 cells. Significance was calculated using an unpaired two-tailed Student's *t*-test of three independent biological replicates. ***$P<0.001$.

expression of defense genes (Hosiner et al., 2009; Thorsen et al., 2007; Wysocki et al., 2004), whereas the transcription factor Sfp1 exits the nucleus upon As(III) exposure, which results in downregulation of protein biosynthesis-related genes (Hosiner et al., 2009). Thus, cells rely on functional nucleocytoplasmic transport to mount an appropriate response to arsenic stress. Nuclear accumulation of chromosomally integrated Yap1-GFP and Msn2-GFP, as well as nuclear exit of Sfp1-GFP were efficient within 5-15 min of As(III) exposure (Fig. S5), consistent with their role as As(III)-responsive factors. These data suggest that nuclear protein import and export is not affected during short-term exposure, possibly because the arsenic that enters cells is first recognized by arsenic-specific sensing and signaling systems before sufficient arsenic accumulates to poison cellular functions. Alternatively, these stress-responsive transcription factors might cross the NE through As(III)-insensitive pathways. Collectively, our data suggest that nuclear transport is disrupted during long-term As(III) stress but remains functional in the initial phase of exposure.

Our chemical-genetic and genetic interaction data (Fig. S1) suggest that arsenic-binding to proteins mediating nucleocytoplasmic transport causes toxicity. In further support of this notion, cells with weakened (temperature-sensitive) alleles of Kap95 (*kap95*-L63A) and Kap121 (*kap121-Δ34*) (Li et al., 2011) were As(III) sensitive (Fig. 7). Thus, disruption of nucleocytoplasmic transport may result in arsenic sensitivity.

## DISCUSSION

This current study implicates nucleocytoplasmic transport as an important target of arsenic toxicity. First, our proteome-wide approach identified 174 arsenic-binding proteins *in vivo*, of which proteins involved in nucleocytoplasmic transport were remarkably enriched. In fact, arsenic bound to most of the importins present in *S. cerevisiae* and we verified arsenic binding to selected importins, exportins and Nups. Second, we demonstrated that importins and Nups mislocalized and aggregated, and that the number of NPCs was reduced in As(III)-exposed cells. Third, we provided evidence that nucleocytoplasmic transport is impaired during As(III) exposure and that cells with defective nuclear protein transport function are As(III) sensitive. Together, our data are consistent with a model in which arsenic-binding to nuclear transport factors leads to their mislocalization and aggregation, disrupting nucleocytoplasmic transport and causing As(III) sensitivity.

Previous *in vivo* proteome-wide studies using the As–biotin probe and various human cell lines typically identified 40-50 candidate arsenic-binding proteins (Dong et al., 2022; Yan et al., 2016; Zhang et al., 2015b, 2007). Our current study yielded 174 proteins representing, to our knowledge, the largest set of *in vivo* arsenic-binding targets reported to date. Some proteins known to bind to arsenic were absent from our hit list, including the As(III)-sensing transcription factor Yap8 (Kumar et al., 2016). Low-abundance proteins are less represented in our dataset, and some true targets might be missed due to the stringent filtering criteria used. A study that applied As–biotin to a human proteome microarray identified 360 candidate arsenic-binding proteins (Zhang et al., 2015a). It is unclear whether proteins are properly folded on the microarray, which is an important aspect because non-native structures could result in cysteine residues being more easily accessible for arsenic binding than in folded native proteins. Nevertheless, the authors of that study demonstrated that one of the 360 candidates, hexokinase 2, is a direct target of arsenic inhibition *in vivo* (Zhang et al., 2015a). Collectively, these studies demonstrate the great utility of the As–biotin probe for proteome-wide identification of binding and toxicity targets, and together provide a valuable resource for mechanistic studies on the toxicity, pathology and therapeutic effects of arsenicals.

While As–biotin is a valuable tool, it cannot discriminate between As(III)- and MAs(III)-binding proteins (Lee and Levin, 2019). By using *mtq2Δ* cells and pretreatments with As(III) and MAs(III) as blocking agents, we identified proteins that bound to As(III) or MAs(III), or to both arsenicals (Table S2). For example, Kap121 and Crm1 bound to both arsenicals, while Kap95 primarily bound to MAs(III) (Fig. 2C). Curiously, the addition of As(III) resulted in more Kap95-GFP being pulled-down by As–biotin (Fig. 2C). While the underlying reason is unknown, we speculate that arsenic-binding to one set of cysteines in Kap95 may lead to a conformational change that exposes a different set of cysteines within Kap95 allowing more As–biotin to bind. Interestingly,

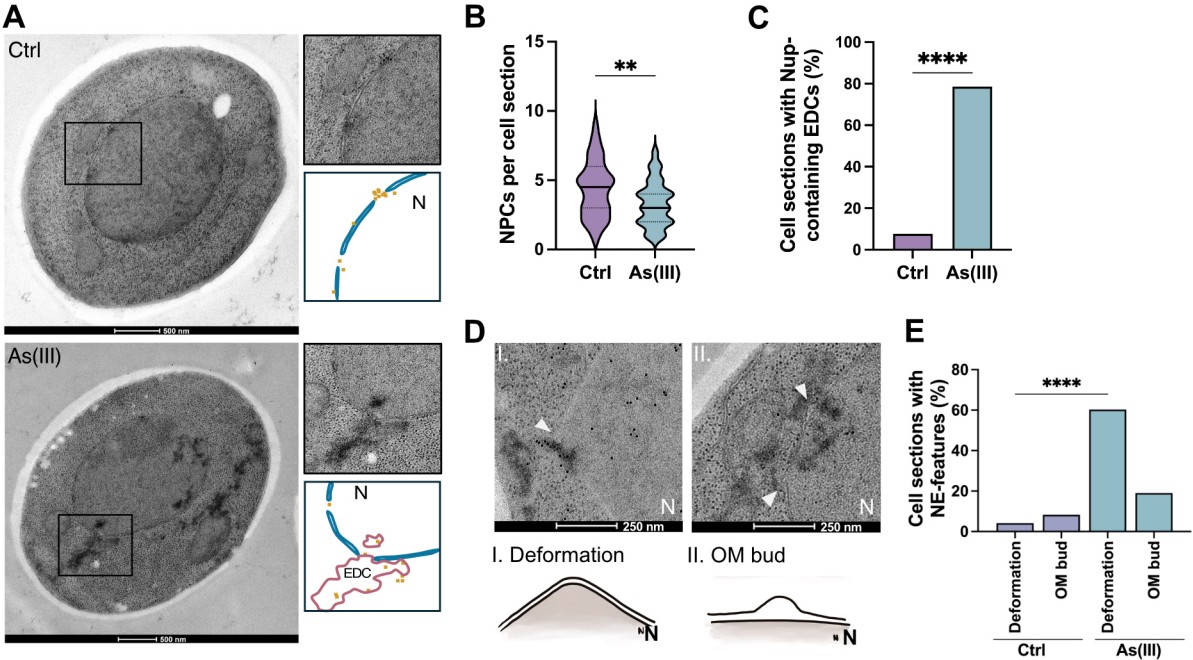

**Fig. 5. As(III) affects nuclear envelope morphology and NPC numbers.** (A) Representative electron micrographs of yeast cells before (Ctrl) and after exposure to 1.5 mM As(III) for 1 h. The rectangles indicate the areas showing NPCs that appear as holes in the nuclear envelope (NE) bilayer (right). Nup localization was assessed with a gold-labelled anti-Nup antibody (Mab414). Protein aggregates appear as electron-dense content (EDC), both in the nucleus and in areas that are free of ribosomes in the cytosol (Panagaki et al., 2021). A model is drawn below each micrograph to visualize the locations of gold particles (in yellow) and EDC (in red). (B,C) Quantification of the number of NPCs per cell section (B) and of the fraction of Nups associated with EDCs/ protein aggregates (C) in unexposed (Ctrl) and As(III)-exposed cells (1.5 mM, 1 h). Nups were detected using a gold-labelled anti-Nup antibody (Mab414). (B) The solid line represents the mean and the dotted lines represent the 25th and 75th percentiles. (D) Representative electron micrographs of yeast nuclei of cells exposed to 1.5 mM As(III) for 1 h showing NE deformations (arrowheads) in which the NE extends into the cytoplasm (I) and where the inner and outer leaflets are separated with the outer leaflet extending into the cytoplasm (II), forming an outer membrane bud (OM bud), often near EDCs/protein aggregates. A model is drawn below each micrograph to visualize the two types of NE deformation. (E) Quantification of the fraction of NE deformations and outer membrane buds (OM bud) per cell section in unexposed (Ctrl) and As(III)-exposed cells (1.5 mM, 1 h). The number of cell sections assessed per condition was 62 for control and 74 for As(III)-exposed cells. Significance was calculated using an unpaired $t$-test. **$P<0.01$ and ****$P<0.0001$. N, nucleus; EDC, electron-dense content, OM bud, outer membrane bud.

previous studies indicated that As(III) and MAs(III) can bind to distinct sets of cysteine thiols in target proteins, thereby eliciting stress-specific responses (Lee and Levin, 2022). How specific cysteine residues distinguish between As(III) and MAs(III) is not known. While our study suggests that the majority of the 174 arsenic-binding proteins can bind to As(III) as well as MAs(III), the exact form of arsenic, the residues involved and the physiological consequences of the binding remain to be defined.

The observations that As(III)-induced Kap95-GFP and Nup84-GFP foci formation were unaffected in the presence of CHX (Fig. 3B; Fig. S4) suggest that arsenic targets the native (folded) form of these proteins promoting their mislocalization and aggregation (Figs 3–5). Previously, we showed that As(III) treatment of yeast cells resulted in the formation of Hsp104-GFP foci that could be prevented by CHX (Andersson et al., 2021; Hua et al., 2022; Ibstedt et al., 2014; Jacobson et al., 2012). Based on this and other findings, we concluded that As(III) primarily targets non-native proteins for misfolding and aggregation *in vivo* (Andersson et al., 2021; Hua et al., 2022; Ibstedt et al., 2014; Jacobson et al., 2012). It is important to note that there is a fundamental difference between Hsp104-GFP foci and Kap95-GFP foci that form in As(III)-exposed cells. Hsp104 is a disaggregase that associates with and reactivates aggregated proteins in *S. cerevisiae* (Glover and Lindquist, 1998) and Hsp104-GFP is a well-established marker for cytosolic protein aggregation (Jacobson et al., 2012; Kaganovich et al., 2008; Panagaki et al., 2021). Thus, As(III)-induced Hsp104-GFP foci formation is a consequence of global aggregation of

cytosolic proteins to which Hsp104 associates while its own activity is unaffected (Hua et al., 2022; Jacobson et al., 2012). In contrast, Kap95-GFP foci represents aggregated (Fig. 3) and probably unfunctional Kap95, since nuclear import of the Kap95/Srp1 cargo GFP-tcNLS was impaired in As(III)-treated cells (Fig. 6). Moreover, unlike Hsp104 and its co-chaperones, karyopherins are unlikely to recognize misfolded proteins. Hence, native and functional Hsp104-GFP forms foci by associating with non-native proteins while Kap95-GFP (and probably also Nup84-GFP) most likely forms foci because it aggregates. We conclude that arsenic can directly modify cysteines in non-native (Hua et al., 2022; Jacobson et al., 2012) as well as native (this work) proteins, driving their unfolding and aggregation, with non-native proteins being particularly vulnerable (Tamás et al., 2014; Wysocki et al., 2023). This raises the question of whether some (or many) of the proteins isolated in the As-biotin experiments are being brought down indirectly in aggregates. However, the As–biotin pull-down experiments would probably not include aggregates because the beads are centrifuged at 1000 $g$, which is not nearly high enough to sediment aggregates. This is also consistent with the relatively low level of overlap between the proteins identified as arsenic binding and those identified by sedimentation of As(III)-induced aggregates (Fig. 1F).

It has been shown that cytosolic protein aggregates can impair nucleocytoplasmic transport by sequestering nuclear shuttle factors (Woerner et al., 2016). Although As(III) induces global aggregation of cytosolic proteins in yeast (Jacobson et al., 2012), our data

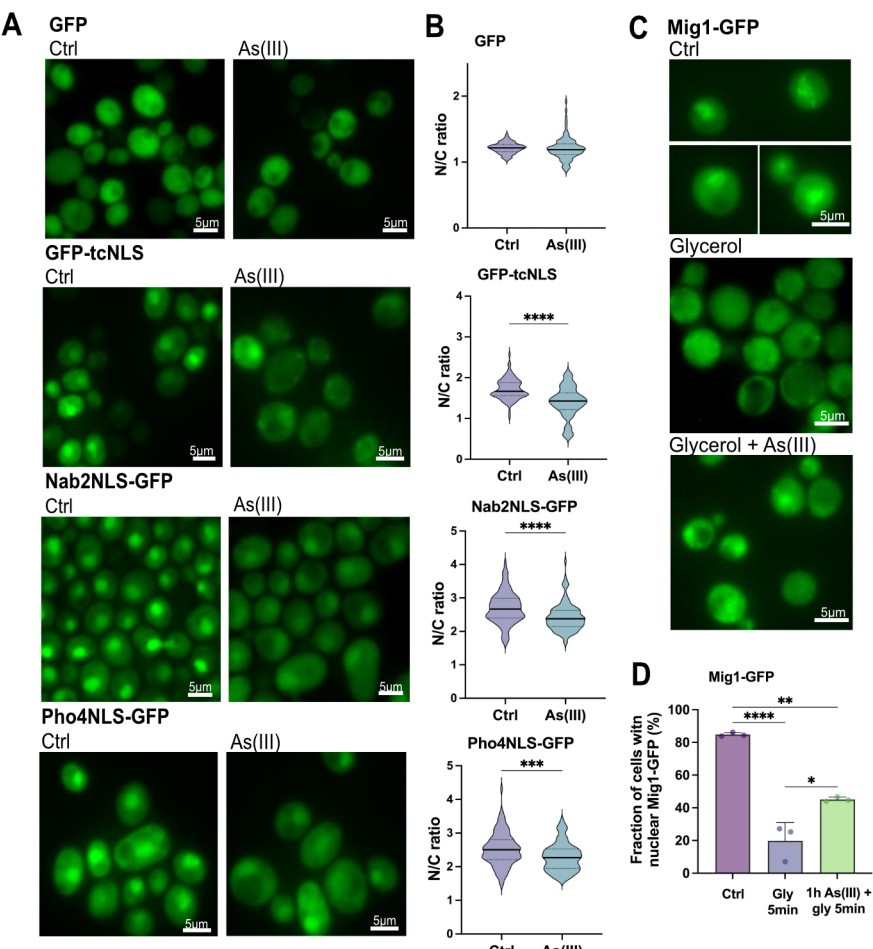

**Fig. 6. Nuclear transport is disrupted during As(III) stress.** (A) Nuclear protein import is inhibited by As(III). Localization of GFP-tagged nuclear transport reporters GFP-tcNLS (recognized by Kap95/Srp1), Nab2NLS-GFP (recognized by Kap104) and Pho4NLS-GFP (recognized by Kap121) was monitored in the absence (Ctrl) or presence of 1.5 mM As(III) for 1 h. GFP without a nuclear sorting sequence was included as a control. Images shown are representative of three biological repeats of 100 cells each. (B) Quantification of the data in A. N/C ratios were calculated by measuring the mean fluorescence intensities in the nucleus (N) and the cytosol (C) in the absence (Ctrl) and presence of 1.5 mM As(III) for 1 h. The graphs show the mean of three biological replicates of around 100 cells per condition measured. The solid line represents the mean and the dotted lines indicate the 25th and 75th percentiles. (C) Nuclear protein export is inhibited by As(III). Mig1-GFP localization was monitored in the presence of glucose and 5 min after a shift to glycerol. Where indicated, cells were pre-incubated with 1.5 mM As(III) for 1 h before the shift to glycerol. (D) Mig1-GFP distribution was scored by fluorescence microscopy and quantified as in Fig. 3B. Data are mean±s.d. of three independent biological repeats of a total of 300 cells. Significance for B and D was calculated using an unpaired *t*-test. \*\**P*<0.01, \*\*\**P*<0.001 and \*\*\*\**P*<0.0001.

point to a more direct mechanism where arsenic impairs nucleocytoplasmic transport by binding to nuclear import and export factors. Arsenic might also affect nucleocytoplasmic transport by binding to Nups, which then leads to their aggregation and a reduced number of NPCs in the NE (Fig. 5). Additionally, As(III) induces morphological abnormalities of the NE with deformations extending into the cytosol (Fig. 5). Interestingly, genetic perturbations that result in NE aberrations, such as NE protrusions that extend into the cytosol, are linked to loss-of-function mutations of NPC components (Thaller and Lusk, 2018), including several Nups identified in our study (e.g. Nup85, Nup188 and Nup145). Thus, the observed NE aberration during As(III) stress may be a direct consequence of arsenic binding to Nups, which interferes with their function.

How relevant are our findings in yeast for understanding toxicity mechanisms and disease processes in humans? The arsenic

concentrations reported in epidemiological studies to cause adverse health effects in humans are in the micromolar range (1–10 μM) (Chen and Costa, 2021; Wysocki et al., 2023). Yeast cells are more resistant to arsenic than mammalian cells, primarily due to efficient detoxification systems that are absent in mammalian cells, such as the As(III) exporter Acr3 and the ABC transporter Ycf1 that transports As(III)-glutathione conjugates into vacuoles (Ghosh et al., 1999; Wysocki et al., 2023). Wild-type yeast cells can grow in the presence of millimolar concentrations of As(III) whereas mutants lacking both Acr3 and Ycf1 are sensitive already at micromolar concentrations (Ghosh et al., 1999), which is in the same range that causes toxicity in mammalian cells (Chen and Costa, 2021; Wysocki et al., 2023). Our previous work indicated that the intracellular arsenic concentration in yeast cells reaches about 10 μM after 1 h of exposure to 1 mM As(III) (Rodrigues et al., 2023), which is comparable to the arsenic concentrations reported in

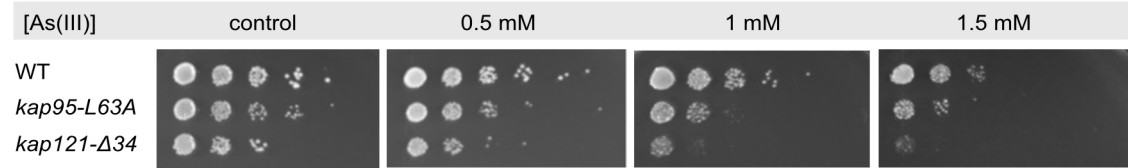

**Fig. 7. Cells defective in nuclear transport function are As(III) sensitive.** Yeast cells that carry a weakened (temperature-sensitive) allele of Kap95 (*kap95-L63A*) or Kap121 (*kap121-Δ34*) were grown to log phase, adjusted to the same optical density, serially diluted and plated onto YNB (yeast nitrogen base) medium with 2% glucose as carbon source and the indicated As(III) concentrations. Plates were incubated at 30°C for 3 days. Images shown are representative of three biological replicates.

epidemiological studies (1-10 µM) to cause adverse health effects in humans (Wysocki et al., 2023). Thus, toxicity targets and mechanisms identified in yeast may be relevant also in human cells. Indeed, the known or predicted 3D structures of the human orthologues of yeast Kap95 (KPNB1), Kap121 (IPO5 and RANBP6), Nup84 (NUP107), and Nup188 (NUP188) all contain proximal cysteines that could potentially serve as binding sites for As(III) and/or MAs(III) in their native folded structures (Fig. S6), and KPNB1 has been identified as a candidate arsenic-binding protein in MCF-7 cells (Zhang et al., 2007). Several studies have linked arsenic exposure to an increased prevalence of neurodegenerative disorders such as Alzheimer's and Parkinson's disease (Rahman et al., 2021; Wysocki et al., 2023). These diseases are characterized by the pathological accumulation of protein aggregates (Soto and Pritzkow, 2018), and there is growing evidence that arsenic contributes to these diseases by impairing protein folding in cells (Wysocki et al., 2023). Interestingly, mislocalization of nucleocytoplasmic transport factors, such as Nups and importins, and the disruption of nucleocytoplasmic transport have been implicated in the pathology of neurodegenerative disorders (Hutten and Dormann, 2020; Kim and Taylor, 2017). We propose that impairment of nucleocytoplasmic transport by arsenic, via direct binding, mislocalization and aggregation of individual nuclear transport factors, as described in this current study in yeast, may be an additional mechanism by which this metalloid contributes to pathology in humans.

To conclude, our study provides previously unreported insights into the molecular mechanisms by which arsenic disrupts cellular function, specifically its impact on nucleocytoplasmic transport. These findings have broad implications for understanding how environmental poisons affect cells and organisms, and may serve as a basis for future research on the toxic and therapeutic effects of arsenicals.

## MATERIALS AND METHODS
### Yeast strains, plasmids and culturing conditions
Yeast strains and plasmids used in this work are listed in Table S3. The *S. cerevisiae* strains are based on BY4741 (Brachmann et al., 1998), the deletion collection (Giaever et al., 2002), the collection of temperature-sensitive mutants of essential yeast genes (Li et al., 2011) and the GFP collection (Huh et al., 2003). The strains harboring nuclear transport reporters have been described previously (Rempel et al., 2019). Plasmids containing HA-tagged versions of Srp1, Kap123, Sxm1, Nup84 and Nup188 were constructed via Gateway Recombination Cloning (Thermo Fisher Scientific) according to the manufacturer's instructions. Gene sequences were amplified by PCR using genomic DNA as template, inserted into the donor plasmid pDONR221 (Thermo Fisher Scientific) and then into the destination vector pAG426GPD-ccdB-HA (Addgene plasmid #14252; deposited by Alberti et al., 2007). All plasmids were verified by sequencing. Yeast strains and plasmids used in this study are available upon request.

Cells were routinely grown at 30°C in minimal YNB (yeast nitrogen base) medium with 2% glucose as a carbon source. Cells containing nuclear transport reporters were grown with 2% raffinose as a carbon source and expression of the reporters was induced with 0.5% galactose for 4 h. Growth on plates was monitored for 2-3 days at 30°C, as previously described (Rodrigues et al., 2023). Where indicated, the following chemicals were added: sodium arsenite (NaAsO₂, S7400) and cycloheximide (C7698) (both from Sigma-Aldrich), biotinyl p-aminophenyl arsenic acid (As-biotin, B394970; Toronto Research Chemicals), and monomethylarsonous acid (CH₅AsO₂, M565100; LGC Standards).

### Arsenic-binding assays
Proteome-wide identification of arsenic-binding proteins was performed with *mtq2Δ* cells grown to early log phase in YNB medium. For the last hour, the medium was switched to YEPD (Yeast Extract Peptone Dextrose)

with 2% glucose, and split into control and treatment cultures. The cultures were pretreated for 10 min with either 1 mM As(III) or 500 µM MAs(III) as blocking agents, followed by a 10 min incubation with 50 µM As-biotin at 30°C. The cultures were collected, pelleted and frozen at −80°C or lysed directly. Cell lysis was performed by bead beating (1 min, 4°C) in immunoprecipitation (IP) buffer [1×TNT buffer (50 mM Tris-HCl at pH 7.5, 150 mM NaCl and 0.5% Triton X-100 at pH 7.5-7.5), 1× protease inhibitor (Complete mini, EDTA-free; Roche Diagnostics), 1×phosphatase inhibitor (PhosSTOP Easypack; Roche Diagnostics) and 1 mM phenylmethylsulfonyl fluoride (PMSF)]. Streptavidin agarose beads (Thermo Fisher Scientific, 20353) were first prepared by washing in 1×TNT buffer and then aliquoted to all lysates. After a 1 h incubation (4°C, rotating), pull-down was performed by centrifugation (1000 *g* for 1 min) followed by washing three times in 1×TNT. Proteins present in the pull-down were separated by SDS-PAGE and identified by LC-MS/MS at the Taplin Mass Spectrometry Facility at Harvard Medical School. To validate arsenic binding to selected proteins, we used wild-type cells carrying plasmids with HA-tagged versions of the corresponding genes or cells that harbored GFP-tagged versions of the genes in their genomes. After As–biotin pull-down and SDS-PAGE, as described above, the proteins were visualized by western blot using anti-GFP rabbit IgG (1:8000, A11122; Invitrogen) or anti-HA mouse IgG (1:1000, sc-7392; Santa Cruz Biotechnologies) primary antibodies, and anti-rabbit IgG (1:3000, 84546) and anti-mouse IgG (1:5000, 84545) secondary antibodies (both from Invitrogen). Unprocessed western blot images are shown in Fig. S7. A more extensive description of the western blot protocol is provided in the section 'Protein aggregate isolation and western blotting'.

### Bioinformatics and protein structure analyses
Negative genetic interactors (including negative genetic, synthetic growth defect and synthetic lethality) of selected arsenic-binding hits were retrieved from the *Saccharomyces* genome database (SGD) (Wong et al., 2023) and compared to a compendium of 712 As(III) sensitive *S. cerevisiae* mutants that contains the genes identified at least once in four previous genome-wide phenotypic screens (Haugen et al., 2004; Pan et al., 2010; Thorsen et al., 2009; Zhou et al., 2009). The significance of the overlaps between datasets was calculated by the hyper-geometric test. All datasets used for comparisons are listed in Table S4. For protein structure analyses, protein sequences were retrieved from SGD (yeast) and UniProt (mammalian) (UniProt, 2025). PDB files were retrieved from the AlphaFold protein structure database (Jumper et al., 2021) and visualized using UCSF ChimeraX version: 1.6.1 (Goddard et al., 2018).

### Fluorescence microscopy and image analyses
Cells expressing GFP-tagged proteins were grown until mid-log phase and either left untreated or exposed to 1.5 mM As(III). Where indicated, 0.2 mg/ml CHX was added at the same time as the As(III). To induce nuclear export of Mig1-GFP, cells were first grown in glucose-containing medium and then shifted to medium containing 2% glycerol. Where indicated, cells were pre-treated with 1.5 mM As(III) for 1 h. All samples were fixed in 3.7% formaldehyde (at room temperature for 30 min) followed by two washes in 1× PBS. Nuclear staining was carried out by incubating fixed cells in ethanol (room temperature for 40 min), washing in 1×PBS and resuspending in 4′,6-diamidino-2-phenylindole (DAPI) solution (D1306; ThermoFisher Scientific). Fluorescent signals were detected using a Zeiss Axiovert 200M fluorescence microscope equipped with Plan-Apochromat 1.40 objectives and appropriate fluorescence light filter sets. Images were taken with a digital camera (AxioCamMR3). The Zeiss ZEN PRO software was used to capture the images and the ImageJ-Fiji software (Schindelin et al., 2012) for quantifications. To avoid bias during manual inspection, two different persons independently quantified NE localization and cytosolic foci.

The steady-state localization of nuclear transport reporters (GFP-tcNLS, Nab2NLS-GFP, Pho4NLS-GFP and GFP) was determined largely as described previously (Barrientos et al., 2023; Rempel et al., 2019). Cells were grown to mid-log phase in YNB medium containing 2% raffinose and expression of the reporters was induced by adding 0.5% galactose for 4 h followed by the addition of 1.5 mM As(III) for 1 h. N/C ratios were quantified by measuring the mean fluorescence intensity in the nucleus and

in the cytosol. The nucleus was outlined along the NE using Nup49-mCherry. Care was taken to exclude the vacuole when choosing a field in the cytosol. All measured values were corrected for background fluorescence and the ratio of nuclear versus cytosolic signal (N/C ratio) was calculated for three replicates and averaged. To avoid bias, the images for quantifying the N/C ratios were anonymized for the control and treatment samples.

## Protein aggregate isolation and western blotting

Protein aggregates were isolated as described previously (Hua et al., 2022; Weids and Grant, 2014). Cells were grown to mid-log phase, unexposed or exposed to 1.5 mM As(III) for 1 h, collected by centrifugation at 5000 **g**, resuspended in lysis buffer [50 mM potassium phosphate buffer at pH 7, 1 mM EDTA, 5% glycerol, 1 mM PMSF, EDTA-free protease inhibitor cocktail (Roche Diagnostics)], and lysed with 2.5 mg/ml lyticase (Sigma-Aldrich) for 30 min at 30°C. Cells were disrupted using sonication on ice (Sonifier 150, Branson Ultrasonics; 8×5 s pulses, 50% amplitude) and the total lysates collected by centrifugation. Protein concentrations in the lysates were adjusted to equal for all samples. Aggregated proteins were isolated by centrifugation of the lysates, resuspended in lysis buffer containing 20% NP40 twice, followed by washing and resuspension of the pellet in lysis buffer. The final resuspension was aided by sonication (2×5 s, 50% amplitude on ice). 5×SDS loading buffer (4% SDS, 250 mM Tris Buffer at pH 6.8, 16% β-mercaptoethanol, 30% glycerol and Bromophenol Blue) was added to all samples followed by boiling for 5 min at 95°C. Proteins were separated on a Criterion 4-20% stain free precast gel (Bio-Rad Laboratories) and visualized using a ChemiDoc XRS+ system with UV-activation (Bio-Rad Laboratories). For western blot analysis, proteins were transferred to a PVDF membrane using TransBlot Turbo transfer system (semi-dry transfer, Bio-Rad Laboratories). Membranes were blocked with 5% bovine serum albumin (BSA) in Tris-buffered saline (TBS) containing 0.05% Tween 20 (TBS-T) for 1 h at room temperature followed by overnight incubation at 4°C with an anti-GFP antibody (1:8000, A11122; Invitrogen). Membranes were washed three times with TBS-T, incubated for 2 h with anti-rabbit-IgG secondary antibody (1:5000, 84546; Invitrogen) and washed with TBS-T followed by signal detection using the ChemiDoc XRS+ imaging system (Bio-Rad Laboratories). Band intensities were quantified by ImageJ and corrected for local background. Pulldown-to-input ratios were calculated for each sample. The ratio for the As-biotin sample was set to 1, and relative values for samples pretreated with As(III) or MAs(III) were obtained by dividing their ratios by the As–biotin ratio. Unprocessed western blot images are shown in Fig. S7.

## Immunoelectron microscopy

Immuno-EM was performed largely as described previously (Panagaki et al., 2021). Cells were grown to early log phase and either left untreated or exposed to 1.5 mM As(III) for 1 h. Samples were high-pressure frozen (Wohlwend HPF Compact 3, Sennwald, Switzerland) followed by freeze substitution in 2% uranyl acetate (UA) dissolved in acetone, and embedded into HM20 lowicryl resin (Polysciences) that was UV polymerized at −50°C. The resin was sectioned in 70 nm sections and placed on mesh grids. The sections were fixed in 1% paraformaldehyde in PBS for 10 min and blocked with 0.1% fish skin gelatine and 0.8% BSA in PBS for 1 h. For detection of Nups, samples were incubated for 2 h with 1:120 dilution of the mouse monoclonal anti-NPC antibody Mab414 (Abcam, ab24609) at 4°C, followed by incubations at room temperature with 1:150 dilution of rabbit anti-mouse immunoglobulin (Agilent/Dako, E0433) for 1 h and with 1:70 diluted 10 nm gold-conjugated protein A antibody (CMC UMC Utrecht, The Netherlands) for 30 min. Glutaraldehyde (2.5%) was applied to sections for 1 h, followed by contrast staining in 2% UA for 5 min and 1 min in Reynold's lead citrate (Reynolds, 1963). Three washing steps (20 min, PBS) were carried out after incubation with each antibody. Images were acquired at 120 kV on a Tecnai T12 transmission electron microscope equipped with a Ceta CMOS 16 M camera (Thermo Fisher Scientific). Quantifications were made using IMOD (Kremer et al., 1996) and statistics with GraphPad Prism 10.

## Statistical information

Depending on the comparisons made, significance was calculated using an unpaired two-tailed Student's *t*-test or the hyper-geometric test, as described in the text and figure legends (*$P<0.05$, **$P<0.01$, ***$P<0.001$ and ****$P<0.0001$). The graphs show the mean±s.d. and the number of independent biological repeats. The number of cells or cell sections assessed are indicated in the figure legends.

### Acknowledgements

We thank Liesbeth M. Veenhoff (University of Groningen, The Netherlands) for providing yeast strains. pAG426GPD-ccdB-HA was a gift from Susan Lindquist (Addgene plasmid #14252). We acknowledge the Taplin Mass Spectrometry Facility at Harvard Medical School for protein identification by LC-MS/MS.

### Competing interests

The authors declare no competing or financial interests.

### Author contributions

Conceptualization: J.L.H., D.E.L., M.J.T.; Formal analysis: E.L., J.L., J.L.H., D.E.L., M.J.T.; Funding acquisition: E.L., J.L.H., D.E.L., M.J.T.; Investigation: E.L., J.L., J.M., K.K., N.K., C.G., R.M.; Project administration: M.J.T.; Supervision: J.L.H., D.E.L., M.J.T.; Visualization: E.L., J.L.H., M.J.T.; Writing – original draft: E.L., M.J.T.; Writing – review & editing: E.L., J.L., J.M., K.K., N.K., C.G., R.M., J.L.H., D.E.L., M.J.T.

### Funding

This work was supported by grants from the National Institutes of Health (R01GM138413 and R01GM48533 to D.E.L.), Cancerfonden (211-865 to J.L.H.), the Vetenskapsrådet (2019-4004 to J.L.H. and 2018-03577 to M.J.T.), the Stiftelsen Bengt Lundqvists Minne, the Sven och Dagmar Saléns Stiftelse (to E.L.) and the Olle Engkvists Stiftelse (216-0452) to M.J.T. Open Access funding provided by University of Gothenburg. Deposited in PMC for immediate release.

### Data and resource availability

All data supporting the findings of this study are contained within the article and the supplementary information.

### Peer review history

The peer review history is available online at https://journals.biologists.com/jcs/lookup/doi/10.1242/jcs.263889.reviewer-comments.pdf

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
