## [Peer Review File · Journal of Cell Science]

Arsenic binds to nuclear transport factors and disrupts nucleocytoplasmic transport

Emma Lorentzon, Jongmin Lee, Jakub Masaryk, Katharina Keuenhof, Nora Karlsson, Charlotte Galipaud, Rebecca Madsen, Johanna L. Höög, David E. Levin and Markus J. Tamás

DOI: 10.1242/jcs.263889

Editor: Megan King

Review timeline

Original submission:	31 January 2025
Editorial decision:	24 February 2025
First revision received:	18 June 2025
Accepted:	20 July 2025

Original submission

First decision letter

MS ID#: jcs.263889

MS TITLE: Direct binding of arsenicals to nuclear transport factors disrupts nucleocytoplasmic transport

AUTHORS: Emma Lorentzon; Jongmin Lee; Jakub Masaryk; Katharina Keuenhof; Nora Karlsson; Charlotte Galipaud; Rebecca Madsen; Johanna L Höög; David E Levin; Markus J Tamás

ARTICLE TYPE: Research Article

Dear Dr Tamás,

We have now reached a decision on the above manuscript.

To see the reviewers' reports and a copy of this decision letter, please go to:

As you will see, the reviewers raise a number of substantial criticisms that prevent me from accepting the paper at this stage. They suggest, however, that a revised version might prove acceptable, if you can address their concerns. If you think that you can deal satisfactorily with the criticisms on revision, I would be pleased to see a revised manuscript. We would then return it to the reviewers. Of note, please ensure that you communicate clearly in your response to the reviewers those aspects that relate to the standards for the budding yeast model, particularly arsenic concentrations and the availability of antibodies.

Reviewer 1

In their manuscript "Direct binding of arsenicals to nuclear transport factors disrupt nucleocytoplasmic transport", Lorentzon and coworkers identify several nuclear transport receptors (importins and exportins) and proteins of the nuclear pore complex (nucleoporins) as prominent arsenic-binding proteins. The authors then describe effects of As(III) on the localization of such proteins, on NPC/NE-morphology and, with respect to cellular functions, on different nuclear transport pathways.

A major concern here is the concentration of As(III) that is used in different assays. Typically, the authors use millimolar concentrations to observe effects (e.g. on nucleocytoplasmic transport, Fig. 6). In other studies (enzyme assays, whole cell assays etc.; Ramadan et al. 2009 (doi: 10.1021/bi801988x); Hofmann et al. 2020 (DOI: 10.1002/advx.201902130) Zhang et al. 2010 (DOI: 10.1126/science.1183424) low micromolar concentrations were used, even in preincubation steps. The question arises, how specific the observed effects in yeast really are, in particular in light of the rather modest effects that are seen in functional assays (again, on nucleocytoplasmic transport, Fig. 6). Obviously, high concentrations are needed to affect cell growth in yeast (Fig. 7). Is there a way to estimate the intracellular concentration of As(III)? The authors could try a simple system in higher eukaryotic cells (e.g. HeLa cells) to monitor possible effects of arsenic on nuclear transport (at micromolar concentrations that are known to affect cellular functions).

Major points:

The authors suggest that cysteine residues (or cysteine pairs) are responsible for As-interactions (as has been shown in other studies). They could express either one of the importins with or without the relevant cysteines as suggested in Fig. S2 in bacteria and test them for binding to immobilized As-biotin. Functionally, mutant proteins lacking such cysteines should rescue As(III) effects on nuclear transport in yeast.

Figure 6: Pictures of treated and untreated cells must be shown to get a better idea about the effects. The level of nuclear accumulation for GFP-tcNLS seems rather low, in particular compared to GFP alone. The authors could try double- or triple-GFP, which should be largely excluded from the nucleus (and not diffuse back into the cytoplasm). NLS-versions would then allow a better assessment of active nuclear import - and of possible As(III)-effects on transport.

Minor points:

Figure 2: Why do the authors use tagged versions of the individual proteins? Nuclear transport receptors are abundant proteins and should be detectable using specific antibodies. For blots, the molecular weight must be indicated. Also, the amount of material in the input should be mentioned (what is the percentage of protein that actually interacts with immobilized As-biotin?). Results should be quantified, as the observed differences are rather small. Why does the level of Kap95-GFP go up in 2C in the presence of As(III)? Have the authors tried to preincubate a cell lysate (instead of intact cells) with competing As(III) or MAs(III)?

Legend (typo): "Cells were pretreated..."

Figure 3: Weight markers should be included, as above. Again, what is the percentage in "Total"? Are there any co-aggregates of different importins or exportins (or nucleoporins)?

Figure 4: What is the earliest time point for cytoplasmic foci to appear (as there is no big difference between 1 and 5 h of treatment)?

Figure S4B: Molecular weight markers are missing. Which antibodies were used for detection? The quality of the blots is not convincing. Levels of endogenous nucleoporins should be analyzed here.

Figure S5: The title does not really fit, as these factors (Yap1-GFP etc.) are affected by As(III). To what extent do the reporter proteins accumulate in the nucleus? Pictures of cells should be shown as well.

Together, the identified interaction of transport factors with As(III) is interesting. Particularly in light of the high As(III) concentrations, it remains unclear, however, if the observed (small) effects on nuclear transport are a direct consequence of binding of arsenic compounds to the identified transport factors (as suggested in the title) or if they arise via indirect effects e.g. on transcription or a general stress response.

Reviewer 2

SUMMARY OF THE ADVANCE MADE IN THIS PAPER AND ITS POTENTIAL SIGNIFICANCE TO THE FIELD

Arsenic, found in groundwater e.g., poses a health threat to humans. It is not fully understood which proteins are most sensitive to arsenic and why. The work by Emma Lorentzon and coauthors identifies proteins from the nuclear transport machinery as targets of arsenic in baker's yeast. They perform a proteomics-based screen and identify 174 candidate arsenic binding proteins. Compared to previous studies this represents a large set of in vivo arsenic-binding proteins. Analyzing the hits, they find confirmation for a preference of As(III)/MAs(III) for cysteine residues in proteins in vivo. The paper performs a number of bioinformatics analysis to provide confidence in the dataset. E.g., and overlap with a previous yeast HIP study and genetic interactions. They observed significant enrichments in As(III) sensitivity among negative genetic interactors of selected arsenic-binding protein. There was also significant overlap with proteins that aggregated in As(III)-exposed yeast cells suggesting that arsenic binding induces aggregation. Interestingly, the arsenic-binding proteins were enriched for nucleocytoplasmic transport proteins including nups, Kaps and exportins, hence nucleocytoplasmic transport may be at risk. Indeed, the structures of Nups and Kaps identified in the screen have cysteines in pairs and motifs that could be modified. Imaging data shows that Kap95-GFP is indeed mislocalized in cytosolic foci upon arsenic treatment. These foci form independent of translation and do not readily dissolve after removal of the arsenic suggesting they represent aggregated Kap95. Imaging experiments with Nup84-GFP and EM also show mislocalization from the NE and into foci that were often localized at sites of NE deformations extending into the cytoplasm. Arsenic treatment results in a reduction of NPCs at the NPC and transport defects of GFP-tcNLS, Nab2NLS-GFP, Pho4NLS-GFP and Mig1-GFP. While transport is impacted at longer timescales apparently the system is robust early on so that a response to arsenic can be orchestrated. Indeed, the authors show that the transport of Yap1-GFP, Msn2-GFP and Sfp1-GFP, involved in signaling the arsenic toxicity is functional at early timepoints of As(III).

Overall, the study convincingly reports a novel connection between arsenic toxicity and nucleocytoplasmic transport. The work adds in a novel way to an increasing body of work showing that there are close relationships between nuclear transport and ageing and aggregation-related pathologies. The work is well written, the figures are complete and intuitive and the data is relevant and novel. I enthusiastically support publication after a few minor revisions.

SUGGESTIONS TO AUTHORS

Minor comments

- 1) P17 Fluorescence microscopy and image analyses please add that concentrations and exposure times with As(III) are listed in the figure legends. What is the rationale of using 1, mM or 1,5 mM As(III) in the different experiments?
- 2) Figure 3B please indicate how bias was avoided in the manual inspection. Was this analysis done double blind?
- 3) Figure 4B could add an inset to show what the cytosolic foci, "often with the fluorescence signal extending from the NE into the cytosol" look like.
- 4) Fig 5 please clarify OM bud or OM NEB
- 5) Could consider integrating the structures, e.g., S2 Kap95 into main text

First revision

Author response to reviewers' comments

Dear Dr Tamas,

We have now reached a decision on the above manuscript.

To see the reviewers' reports and a copy of this decision letter, please go to: View Reviewer Comments

As you will see, the reviewers raise a number of substantial criticisms that prevent me from accepting the paper at this stage. They suggest, however, that a revised version might prove acceptable, if you can address their concerns. If you think that you can deal satisfactorily with the criticisms on revision, I would be pleased to see a revised manuscript. We would then return it to the reviewers. Of note, please ensure that you communicate clearly in your response to the reviewers those aspects that relate to the standards for the budding yeast model, particularly arsenic concentrations and the availability of antibodies.

Please ensure that you clearly highlight all changes made in the revised manuscript. Please avoid using 'Tracked changes' in Word files as these are lost in PDF conversion.

I should be grateful if you would also provide a point-by-point response detailing how you have dealt with the points raised by the reviewers in the 'Response to Reviewers' box. Please attend to all of the reviewers' comments. If you do not agree with any of their criticisms or suggestions please explain clearly why this is so.

I look forward to receiving your revised manuscript.

With best wishes,

Megan C King

Handling Editor

Dear Dr. King,

Thank you for evaluating the manuscript jcs.263889. We appreciate the reviewers' constructive comments and suggestions and the extension of the revision due date. We now submit a revised manuscript which has been adjusted and improved accordingly and provide a point-to-point answer to the questions raised. Specifically, we address the aspects that relate to the standards for the budding yeast model, particularly arsenic concentrations and the availability of antibodies, as requested. Text changes (additions and deletions) are marked in red in the revised manuscript, clean copies of the manuscript and the supporting information are provided, and text and figures have been adjusted according to JCS' guidelines. We hope that the manuscript will be acceptable in its current revised version.

Kind regards,

Markus Tamas

Comments from the Reviewers:

Reviewer 1: In their manuscript "Direct binding of arsenicals to nuclear transport factors disrupt nucleocytoplasmic transport", Lorentzon and coworkers identify several nuclear transport receptors (importins and exportins) and proteins of the nuclear pore complex (nucleoporins) as prominent arsenic-binding proteins. The authors then describe effects of As(III) on the localization of such proteins, on NPC/NE-morphology and, with respect to cellular functions, on different nuclear transport pathways.

A major concern here is the concentration of As(III) that is used in different assays. Typically, the authors use millimolar concentrations to observe effects (e.g. on nucleocytoplasmic transport, Fig. 6). In other studies (enzyme assays, whole cell assays etc.; Ramadan et al. 2009 (doi: 10.1021/bi801988x); Hofmann et al. 2020 (DOI: 10.1002/adv.201902130)

Zhang et al. 2010 (DOI: 10.1126/science.1183424) low micromolar concentrations were used, even in preincubation steps. The question arises, how specific the observed effects in yeast really are, in particular in light of the rather modest effects that are seen in functional assays (again, on nucleocytoplasmic transport, Fig. 6). Obviously, high concentrations are needed to affect cell growth in yeast (Fig. 7). Is there a way to estimate the intracellular concentration of As(III)? The authors could try a simple system in higher eukaryotic cells (e.g. HeLa cells) to monitor possible effects of arsenic on nuclear transport (at micromolar concentrations that are known to affect cellular functions).

Thank you for the constructive review.

- As(III) concentration used. Compared to mammals, yeast is more resistant to arsenic mainly due to efficient detoxification systems that are absent in mammalian cells (e.g. the As(III) exporter Acr3 and the ABC transporter Ycf1, which sequesters As(III)-glutathione conjugates into the vacuole). Wild type yeast cells can grow in the presence of mM concentrations of As(III) whereas mutants lacking Acr3 and Ycf1 are sensitive already at μM concentrations, which is in the same range as the concentrations that cause toxicity in mammalian cells. We have added this information and the relevant references to the revised manuscript. Note that there are differences in arsenic resistance also between animal species, e.g. Chinese hamster ovary (CHO-K1) cells are about 10 times more resistant to As(III) than human fibroblasts. Also human cell lines can vary greatly in susceptibility to arsenic (reviewed in Wysocki et al 2023 DOI: 10.1007/s00018-023-04992-5).

- Intracellular arsenic concentrations. Our previous work (e.g. Rodrigues et al 2023 DOI: 10.1002/1873-3468.14638) indicates that the intracellular arsenic concentration reaches about 10 μM 1 hour after addition of 1 mM As(III). Thus, exposing yeast cells to mM concentrations of extracellular As(III) results in μM concentrations of intracellular arsenic, which correspond to the arsenic concentrations reported in epidemiological studies (1-10 μM) to cause adverse health effects in humans (reviewed in Wysocki et al 2023 DOI: 10.1007/s00018-023-04992-5) and in the studies pointed out by the reviewer above. We have added this information and the relevant references to the revised manuscript.

- Relevance for higher eukaryotes. We agree with the reviewer that it will be important to elucidate whether arsenic affects nucleocytoplasmic transport also in mammalian cells. However, performing the experiments described in yeast in a different model system will require substantial efforts and is better suited for a future manuscript. Nevertheless, in the revised manuscript, we added a section that discusses the relevance of our findings in yeast for understanding toxicity mechanisms and disease processes in humans. We also include novel data indicating that the known or predicted 3D structures of the human orthologues of Kap95 (KPNB1), Kap121 (IPO5 and RANBP6), Nup84 (NUP107), and Nup188 (NUP188) all contain proximal cysteines that could potentially serve as binding sites for As(III)/MAs(III) in their native folded structures (novel Fig. S6). Thus, toxicity targets and mechanisms identified in yeast may be relevant also in human cells.

Major points:

The authors suggest that cysteine residues (or cysteine pairs) are responsible for As-interactions (as has been shown in other studies). They could express either one of the importins with or without the relevant cysteines as suggested in Fig. S2 in bacteria and test them for binding to immobilized As-biotin. Functionally, mutant proteins lacking such cysteines should rescue As(III) effects on nuclear transport in yeast.

We agree with the reviewer that it is important to pin-point the cysteine residues that bind to arsenic and to use the corresponding mutants to elucidate how arsenic binding affects individual importins and nucleoporins, and how the mutated proteins impact nucleocytoplasmic transport in cells. To properly address arsenic binding, protein (mis)localization and aggregation, and nucleocytoplasmic transport *in vivo*, we would need to generate yeast strains that carry the Cys-mutant versions of these proteins in their genomes together with the appropriate nuclear transport reporters. This approach is complicated by the fact that most importins and nucleoporins are encoded by genes that are essential for viability. Moreover, specific cysteines in karyopherins and Nups have been shown to influence nucleocytoplasmic transport (e.g. Yoshimura et al 2013, DOI: 10.1242/jcs.124172; Zhang et al 2020 DOI: 10.1016/j.celrep.2020.108484; Wing et al DOI: 2022 10.1038/s41580-021-00446-7); thus, there is a possibility that the cysteine mutant versions may alter nuclear transport already in the absence of arsenic. Therefore, although the suggested experiments are very valid, we feel that they will require substantial efforts and are better suited as focus of a future manuscript. As stated in the Discussion section in the original manuscript, ‘...future work is required to elucidate the exact form of arsenic that binds to individual proteins as well as the physiological consequences of this binding.’

Figure 6: Pictures of treated and untreated cells must be shown to get a better idea about the effects. The level of nuclear accumulation for GFP-tcNLS seems rather low, in particular compared

to GFP alone. The authors could try double- or triple-GFP, which should be largely excluded from the nucleus (and not diffuse back into the cytoplasm). NLS-versions would then allow a better assessment of active nuclear import - and of possible As(III)-effects on transport.

- We added representative images of untreated and exposed cells as requested.

- We agree that the steady-state N/C ratio of GFP-tcNLS is somewhat lower than expected. However, we do observe a significant decrease in the N/C ratio when we treat these cells with As(III), which is consistent with less import of the reporter. Quantitatively, the decrease in N/C ratios in response to As(III) exposure in our study is in the same range as in other studies using the same reporters (Rempel et al 2019 DOI: 10.7554/eLife.48186; Semmelink et al 2022 DOI: 10.1101/2022.04.13.488135).

Minor points:

Figure 2: Why do the authors use tagged versions of the individual proteins? Nuclear transport receptors are abundant proteins and should be detectable using specific antibodies. For blots, the molecular weight must be indicated. Also, the amount of material in the input should be mentioned (what is the percentage of protein that actually interacts with immobilized As-biotin?). Results should be quantified, as the observed differences are rather small. Why does the level of Kap95-GFP go up in 2C in the presence of As(III)? Have the authors tried to preincubate a cell lysate (instead of intact cells) with competing As(III) or MAs(III)?

Legend (typo): "Cells were pretreated..."

- Tagged versions of individual proteins. The use of tagged versions of individual proteins is standard in the yeast field as yeast-specific antibodies are often not commercially available whereas genome-wide collections of yeast strains with chromosomal GFP-tags (or other epitope-tags) are easily available.

- The requested info related to the blots has been added. For clarity reasons, we added the molecular size of the proteins to the legend rather than in the figure itself. The original blots including the MW markers are available in the Supplemental material (Fig. S7). Quantifications are shown in the figures.

- We noted that the levels of Kap95 increase in the presence of As(III) but the underlying reason is unknown. One possibility is that arsenic-binding to one set of cysteines in Kap95 may lead to a conformational change which exposes a different set of cysteines within Kap95 allowing more As-biotin to bind. This is now mentioned in the revised manuscript.

- The typo has been corrected.

Figure 3: Weight markers should be included, as above. Again, what is the percentage in "Total"? Are there any co-aggregates of different importins or exportins (or nucleoporins)?

- The requested info has been added. For clarity reasons, we added the molecular size of the proteins to the legend rather than in the figure itself. The original blots including the MW markers are available in the Supplemental material (Fig. S7).

- Co-aggregates. It is not clear to us what the reviewer means with co-aggregates. If the reviewer asks whether other proteins beside the ones tested in this study aggregate in As(III)-exposed cells, the answer is yes; we have previously published a dataset of proteins that aggregate during As(III) exposure in yeast (Jacobson et al 2012, DOI: 10.1242/jcs.107029; Ibstedt et al 2014, DOI: 10.1242/bio.20148938). This dataset has been used in Fig. 1F and the list of proteins are included in Table S4.

Figure 4: What is the earliest time point for cytoplasmic foci to appear (as there is no big difference between 1 and 5 h of treatment)?

The uptake of arsenic into yeast cells is time- and concentration-dependent and our earlier studies have shown that 1 h after external addition of As(III), the intracellular arsenic concentration is

sufficient to affect cellular functions and responses including protein misfolding and aggregation (e.g. Thorsen et al 2006, DOI: 10.1091/mbc.e06-04-0315; Jacobson et al 2012, DOI: 10.1242/jcs.107029; Andersson et al 2021 DOI: 10.1242/jcs.258338). Therefore, we chose the 1h time-point as our earliest time-point. Since there is no difference between the 1h time-point and the later time-points, we removed the latter from Fig. 4.

Figure S4B: Molecular weight markers are missing. Which antibodies were used for detection? The quality of the blots is not convincing. Levels of endogenous nucleoporins should be analyzed here.

The abundance of endogenous nucleoporins before and after As(III) exposure have previously been determined by quantitative proteomics (Guerra-Moreno et al 2015, DOI: 10.1074/jbc.M115.684969), showing that the amounts of Nup84 and Nup188 (as well as all other Nups detected) remain unchanged. As published data support our observations, the blot shown in Fig. S4B does not add new information. We have therefore removed the blot and refer to published data instead in the revised manuscript

Figure S5: The title does not really fit, as these factors (Yap1-GFP etc.) are affected by As(III). To what extent do the reporter proteins accumulate in the nucleus? Pictures of cells should be shown as well.

- We changed the figure title and the corresponding text in the manuscript to ‘Nuclear transport is functional during short-term As(III) exposure’.
- The extent of nuclear accumulation of the transcription factors was quantified and the results are shown in the graphs; the y-axis shows the fraction of cells with nuclear localization of the three proteins (Yap1-GFP, Msn2-GFP, Sfp1-GFP, respectively).
- We have added representative pictures.

Together, the identified interaction of transport factors with As(III) is interesting. Particularly in light of the high As(III) concentrations, it remains unclear, however, if the observed (small) effects on nuclear transport are a direct consequence of binding of arsenic compounds to the identified transport factors (as suggested in the title) or if they arise via indirect effects e.g. on transcription or a general stress response.

As explained above, intracellular arsenic concentrations are in the μM range in our assays. Nevertheless, we acknowledge the reviewer’s concern and have slightly modified the title of the manuscript to better reflect the main findings of our work. To gain additional mechanistic insights, we will direct our future efforts into identifying cysteine residues that bind to arsenic in individual importins and Nups and use the mutants to elucidate how arsenic-binding affects these proteins and their transport cargos.

Reviewer 2: SUMMARY OF THE ADVANCE MADE IN THIS PAPER AND ITS POTENTIAL SIGNIFICANCE TO THE FIELD

Arsenic, found in groundwater e.g., poses a health threat to humans. It is not fully understood which proteins are most sensitive to arsenic and why. The work by Emma Lorentzon and coauthors identifies proteins from the nuclear transport machinery as targets of arsenic in baker’s yeast. They perform a proteomics-based screen and identify 174 candidate arsenic binding proteins. Compared to previous studies this represents a large set of in vivo arsenic-binding proteins. Analyzing the hits, they find confirmation for a preference of As(III)/MAs(III) for cysteine residues in proteins in vivo. The paper performs a number of bioinformatics analysis to provide confidence in the dataset. E.g., and overlap with a previous yeast HIP study and genetic interactions. They observed significant enrichments in As(III) sensitivity among negative genetic interactors of selected arsenic-binding protein. There was also significant overlap with proteins that aggregated in As(III)-exposed yeast cells suggesting that arsenic binding induces aggregation.

Interestingly, the arsenic-binding proteins were enriched for nucleocytoplasmic transport proteins including nups, Kaps and exportins, hence nucleocytoplasmic transport may be at risk. Indeed, the structures of Nups and Kaps identified in the screen have cysteines in pairs and motifs that could be modified. Imaging data shows that Kap95-GFP is indeed mislocalized in cytosolic foci upon arsenic treatment. These foci form independent of translation and do not readily dissolve after removal of

the arsenic suggesting they represent aggregated Kap95. Imaging experiments with Nup84-GFP and EM also show mislocalization from the NE and into foci that were often localized at sites of NE deformations extending into the cytoplasm. Arsenic treatment results in a reduction of NPCs at the NPC and transport defects of GFP-tcNLS, Nab2NLS-GFP, Pho4NLS-GFP and Mig1-GFP. While transport is impacted at longer timescales apparently the system is robust early on so that a response to arsenic can be orchestrated. Indeed, the authors show that the transport of Yap1-GFP, Msn2-GFP and Sfp1-GFP, involved in signaling the arsenic toxicity is functional at early timepoints of As(III).

Overall, the study convincingly reports a novel connection between arsenic toxicity and nucleocytoplasmic transport. The work adds in a novel way to an increasing body of work showing that there are close relationships between nuclear transport and ageing and aggregation-related pathologies. The work is well written, the figures are complete and intuitive and the data is relevant and novel. I enthusiastically support publication after a few minor revisions.

Thank you for the constructive review.

SUGGESTIONS TO AUTHORS

Minor comments

1) P17 Fluorescence microscopy and image analyses please add that concentrations and exposure times with As(III) are listed in the figure legends. What is the rationale of using 1, mM or 1,5 mM As(III) in the different experiments?

- As(III) concentrations and exposure times are now mentioned in the figure legends and the methods section.

- Rationale for using different As(III) concentrations. We consistently exposed cells to 1.5 mM As(III) in all experiments that addressed the consequence of arsenic binding to selected importins and nucleoporins. To address whether arsenic impacts nucleocytoplasmic transport during short-term exposure, we also included exposure to 0.5 mM As(III) (in addition to 1.5 mM) to address time- and dose-responses (Fig. S5). For the proteome-wide identification of arsenic-binding proteins, we pretreated cells with 1 mM As(III) as blocking agent and the same concentration was used to validate arsenic binding to selected hits (Fig. 2). Using 1.5 mM As(III) in this experiment would not change the results; if anything, we expect a somewhat stronger blocking effect when pretreating cells with 1.5 mM As(III) vs 1 mM As(III). For the arsenic-binding assays, we used a lower MAs(III) (500 μ M) than As(III) (1 mM) as blocking agents since MAs(III) is more toxic than As(III). This information is now added to the text.

2) Figure 3B please indicate how bias was avoided in the manual inspection. Was this analysis done double blind?

Two individuals independently quantified NE localization and cytosolic foci in Fig 3B with matching results. The images for quantifying the N/C ratio (Figs. 6A and 6B) were blinded for the control and treatment samples. This information has been added to the revised manuscript.

3) Figure 4B could add an inset to show what the cytosolic foci, "often with the fluorescence signal extending from the NE into the cytosol" look like.

For clarity, we changed this sentence to: Nup84-GFP and Nup188-GFP were visible as cytosolic foci in a substantial fraction of the cells.

4) Fig 5 please clarify OM bud or OM NEB

The typo has been corrected (should be OM bud).

5) Could consider integrating the structures, e.g., S2 Kap95 into main text

We prefer to keep the structures in the supplementary data as we have not yet experimentally verified arsenic-binding to specific cysteine residues.

Second decision letter

MS ID#: jcs.263889R1

MS TITLE: Arsenic binds to nuclear transport factors and disrupts nucleocytoplasmic transport

AUTHORS: Emma Lorentzon; Jongmin Lee; Jakub Masaryk; Katharina Keuenhof; Nora Karlsson; Charlotte Galipaud; Rebecca Madsen; Johanna L Höög; David E Levin; Markus J Tamas
Article Type: Research Article

Dear Dr Tamas,

I am happy to tell you that your manuscript has been accepted for publication in Journal of Cell Science, pending standard publication integrity checks.